# Regeneration following tissue necrosis is mediated by non-apoptotic caspase activity

Jacob W Klemm, Chloe Van Hazel, Robin E Harris*

Arizona State University, Tempe, United States

## eLife Assessment

This manuscript reports **fundamental** discoveries on how necrotic cells contribute to organ regeneration through apoptotic signalling to produce cells with non-lethal apoptotic caspase activity that contribute to the regenerated tissue. These findings will be of broad interest to those who study wound repair and tissue regeneration. The strength of the evidence is **solid** and has been improved in the revised version.

**\*For correspondence:**
Robin.Harris@asu.edu

**Competing interest:** The authors declare that no competing interests exist.

**Abstract** Tissue necrosis is a devastating complication for many human diseases and injuries. Unfortunately, our understanding of necrosis and how it impacts surrounding healthy tissue – an essential consideration when developing effective methods to treat such injuries – has been limited by a lack of robust genetically tractable models. Our lab previously established a method to study necrosis-induced regeneration in the *Drosophila* wing imaginal disc, which revealed a unique phenomenon whereby cells at a distance from the injury upregulate caspase activity in a process called Necrosis-induced Apoptosis (NiA) that is vital for regeneration. Here, we have further investigated this phenomenon, showing that NiA is predominantly associated with the highly regenerative pouch region of the disc, shaped by genetic factors present in the presumptive hinge. Furthermore, we find that a proportion of NiA fail to undergo apoptosis, instead surviving effector caspase activation to persist within the tissue and stimulate reparative proliferation late in regeneration. This proliferation relies on the initiator caspase Dronc, and occurs independent of JNK, ROS or mitogens associated with the previously characterized Apoptosis-induced Proliferation (AiP) mechanism. These data reveal a new means by which non-apoptotic Dronc signaling promotes regenerative proliferation in response to necrotic damage.

## Introduction

Necrosis is the rapid, disordered death of cells characterized by the loss of membrane integrity and release of cytoplasmic contents into the surrounding tissue (*Hajibabaie et al., 2023*). This catastrophic type of cell death can occur in diverse tissues and is central to many human conditions, particularly those related to ischemic injuries. Such conditions include chronic illnesses like diabetes, joint disorders, sickle-cell anemia and other inherited and congenital diseases (*Masi et al., 2007*; *Mulay et al., 2016*; *Karsch-Bluman et al., 2019*; *Tonnus et al., 2021*; *Li et al., 2023*), as well as more acute medical events like strokes, heart attacks, bacterial infections and common traumatic injuries (*Konstantinidis et al., 2012*; *Hakkarainen et al., 2014*; *Bonne and Kadri, 2017*; *Wu et al., 2018*). Even therapeutic interventions, in particular treatments for cancer, can result in this devastating form of damage (*Robertson et al., 2017*; *Nakada et al., 2019*; *Yang et al., 2021*). Unfortunately, current strategies to treat necrosis mainly focus on invasive procedures that are often met with limited success. With

such substantial bearings on human health, it is crucial to better understand the effects of necrosis in disease and injury, particularly in the context of tissue repair and regeneration.

We currently have a limited understanding of how necrosis impacts surrounding healthy tissue during wound healing. Indeed, much of our understanding about how cell death influences tissue repair comes instead from models involving programmed cell death (PCD) like apoptosis (*Hajibabaie et al., 2023*). This highly regulated process can be triggered by intrinsic or extrinsic pathways, both of which lead to the activation of caspases that mediate the controlled disassembly of the cell (*Ashkenazi and Salvesen, 2014*). Studies of PCD in a variety of species have shown that cells undergoing apoptosis can release signaling molecules that are interpreted by surrounding tissues to drive wound healing events, such as tissue remodeling, immune responses, survival and proliferation of surrounding cells (*Tseng et al., 2007*; *Fan and Bergmann, 2008a*; *Chera et al., 2009*; *Bergmann and Steller, 2010*; *Li et al., 2010*; *Pellettieri et al., 2010*; *Ryoo and Bergmann, 2012*; *Vriz et al., 2014*; *Fuchs and Steller, 2015*; *Pérez-Garijo and Steller, 2015*; *Fogarty and Bergmann, 2017*; *Pérez-Garijo, 2018*). For example, the signaling molecules Prostaglandin E2 and Hedgehog are produced by dying hepatocytes to induce regenerative proliferation in the vertebrate liver (*Jung et al., 2010*; *Li et al., 2010*), while apoptotic-deficient mice show both impaired liver regeneration and epidermal recovery after wounding (*Li et al., 2010*). Although mitogenic signaling by apoptotic cells is an established and conserved process, whether similar signaling events occur following necrotic cell death is less clear.

Evidence of apoptotic signaling first originated from studies of the larval wing primordia in *Drosophila* (*Pérez-Garijo et al., 2004*; *Ryoo et al., 2004*). This epithelial tissue has been extensively characterized as a model for growth, development and regeneration, including the role that cell death plays in these processes (*Beira and Paro, 2016*; *Worley and Hariharan, 2022*). Ongoing studies of this model have identified an essential signaling network centered on the highly conserved JNK pathway. JNK activates several major signaling pathways including Hippo and JAK/STAT, which have conserved roles in promoting regeneration across species (*Worley et al., 2012*; *Hariharan and Serras, 2017*; *Fox et al., 2020*; *Worley and Hariharan, 2022*), as well as activating JNK itself via overlapping positive feedback loops. One such feedback loop acts through the initiator caspase Dronc (*Drosophila* Caspase-9), which, independent of its role in apoptosis, is translocated to the cell membrane to activate the release of ROS from the NADPH oxidase Duox (*Amcheslavsky et al., 2018*). ROS attracts hemocytes to further activate JNK signaling in the disc through the release of the TNF ligand Eiger (*Fogarty et al., 2016*; *Diwanji and Bergmann, 2018*). In a related pathway, JNK can also lead to the expression of the *Duox* maturation factor *moladietz* (*mol*), thus activating this feedback loop without Dronc (*Khan et al., 2017*; *Pinal et al., 2018*). An important advance in elucidating this network was the ability to generate 'undead cells', using the baculovirus caspase inhibitor P35 to prevent apoptotic cells from dying (*Hay et al., 1994*). These cells therefore persist, releasing mitogenic factors including Wingless (Wg, Wnt1), Decapentaplegic (Dpp, BMP2/4), Spitz (Spi, EGF), or Hedgehog (Hh; *Huh et al., 2004*; *Pérez-Garijo et al., 2004*; *Ryoo et al., 2004*; *Pérez-Garijo et al., 2005*; *Fan and Bergmann, 2008a*; *Pérez-Garijo et al., 2009*; *Morata et al., 2011*; *Fan et al., 2014*). These signals subsequently promote proliferation of the surrounding cells in a phenomenon known as Apoptosis-induced Proliferation (AiP; *Fan and Bergmann, 2008b*; *Ryoo and Bergmann, 2012*; *Fogarty and Bergmann, 2017*).

By contrast, the genetic events following necrosis are less well explored. Necrosis is characterized by swelling and loss of cellular membrane integrity, with the release of cellular contents into the intercellular space causing a significant inflammatory response (*Festjens et al., 2006*; *D'Arcy, 2019*; *Hajibabaie et al., 2023*). Necrosis is highly variable, occurring as a regulated process, for example necroptosis, or as unregulated, caspase-independent lysis (*Ashkenazi and Salvesen, 2014*; *D'Arcy, 2019*). The factors released from necrotic cells are collectively termed Damage-Associated Molecular Patterns (DAMPs; *Vénéreau et al., 2015*; *Roh and Sohn, 2018*), which are thought to interact with pattern recognition receptors (PRRs) on nearby cells, mostly of the Toll-like receptor (TLR) family (*Ming et al., 2014*; *Gong et al., 2020*). DAMPs are understood to mainly consist of fundamental cellular components like histones, chromatin, and actin (*Vénéreau et al., 2015*; *Gordon et al., 2018*; *Roh and Sohn, 2018*), although specific factors have also been described. For example, High-mobility group box 1 (HMBG1) has been characterized as a DAMP in models of spinal cord, cardiac and muscle injury where it promotes angiogenesis, attracts repair cells and induces proliferation (*Vénéreau et al., 2015*), as well as in *Drosophila* models of necrosis (*Nishida et al., 2024*). However, the overall role of

DAMPs and how they influence healing and regeneration through interaction with healthy tissues has yet to be fully explored.

To better investigate necrosis-induced wound repair and regeneration, our lab developed a method to rapidly and reproducibly induce necrotic cell death within the developing *Drosophila* wing imaginal discs (*Klemm et al., 2021*). Using a genetic ablation system we previously established, named Duration and Location (DUAL) Control (*Harris, 2023*), we can induce necrosis in the wing disc via expression of a leaky cation channel *GluR1^{LC}*(*Liu et al., 2013*; *Yang et al., 2013*). Using this system (*DC^{GluR1}*), we showed that wing discs are capable of fully regenerating following necrotic injury at a rate comparable to that of damage induced by apoptosis (*Klemm et al., 2021*). However, while apoptotic ablation leads to JNK signaling and extensive caspase activity, we found that necrosis leads to only minor levels of JNK-mediated apoptosis, which is confined to the wound edge, but unexpectedly generates significant caspase activity in cells distant from the injury. We called this non-autonomous caspase activation Necrosis-induced Apoptosis (NiA; *Klemm et al., 2021*). Unlike normal apoptotic cells, NiA form entirely independent of JNK signaling, and cannot be made undead using P35. We also demonstrated that NiA is essential for regeneration, although the mechanism was unclear.

Here, we have further characterized the NiA phenomenon, finding that only regeneration-competent areas of the wing disc can produce NiA following damage, in part due to WNT and JAK/STAT signaling in the hinge that limits NiA to the pouch. Building upon our finding that NiA is necessary for regeneration, we show that NiA leads to localized proliferation significantly later in regeneration than previously observed. Using tools to trace caspase activity and cell death, we demonstrate that this is possible because a proportion of NiA survive effector caspase activation and persist late into regeneration where they promote proliferation. Finally, we show that this proliferation relies on the initiator caspase Dronc, but surprisingly does not involve established AiP mechanisms. Our data suggest a model in which necrotic injuries induce caspase activity in cells at a distance from the injury, some of which undergo JNK-independent apoptosis (NiA), while others survive and promote proliferation through a novel non-apoptotic function of Dronc, which is separate from its role in AiP. We refer to these surviving NiA cells as Necrosis-induced Caspase-Positive (NiCP) cells. These findings reveal an important genetic response to lytic cell death that could potentially be leveraged to augment regeneration of necrotic wounds.

## Results

### Formation of NiA occurs primarily in the wing pouch

Previously, we found that NiA occurs in the lateral pouch (LP) upon induction of necrosis in the distal pouch with *DC^{GluR1}* (*Figure 1A, D and E*, yellow arrowheads in E) (*Klemm et al., 2021*). The wing disc itself comprises different identities reflecting the adult structures they ultimately create, including the pouch, hinge and notum, which are themselves divided into compartments; anterior/posterior and dorsal/ventral (*Figure 1B*). Since these various disc identities have distinct regenerative capacities stemming from their different genetic responses to damage (*Martín et al., 2017*), to better understand the formation of NiA and the role it plays in regeneration we tested whether necrosis occurring in different areas of the disc leads to NiA. To do so, we utilized *GAL4/UAS/GAL80^{ts}* to conditionally express *UAS-GluR1^{LC}* (*Liu et al., 2013*; *Yang et al., 2013*) in the pouch, hinge or notum tissues (*Figure 1C*). As an initial test, we attempted to recapitulate our original observations made using *DC^{GluR1}* by employing an enhancer of the *spalt* gene driving *GAL4* (*R85E08^{ts}>GluR1*) to cause necrosis in the distal pouch (*Figure 1F and G*). As anticipated, NiA are formed in the lateral pouch following 20 hr of ablation (denoted as 0 hr, when larvae are downshifted to 18 °C; *Figure 1G*). In this figure and others, NiA are recognized as cells positive for the cleaved caspase cDcp-1 and negative for GFP that labels the ablation domain (*UAS-GFP*; *Figure 1E and G*, yellow arrowheads), which indicates that these caspase-positive cells originate outside the area of ablation. This test confirms that the NiA phenomenon occurs independent of the ablation system used. Notably, NiA are consistently absent from the presumptive hinge region surrounding the pouch following both *R85E08^{ts}>GluR1* or *DC^{GluR1}* ablation (*Figure 1E and G*). Indeed, outside of the pouch, cDcp-1 staining is only observed in a small area of the posterior pleura (*Figure 1E*, red arrowhead) and at low levels stochastically across the disc due to stress induced by temperature changes (*Klemm et al., 2021*). To further investigate the extent to which necrotic pouch tissue can induce NiA we next ablated the entire pouch using *rotund-GAL4*

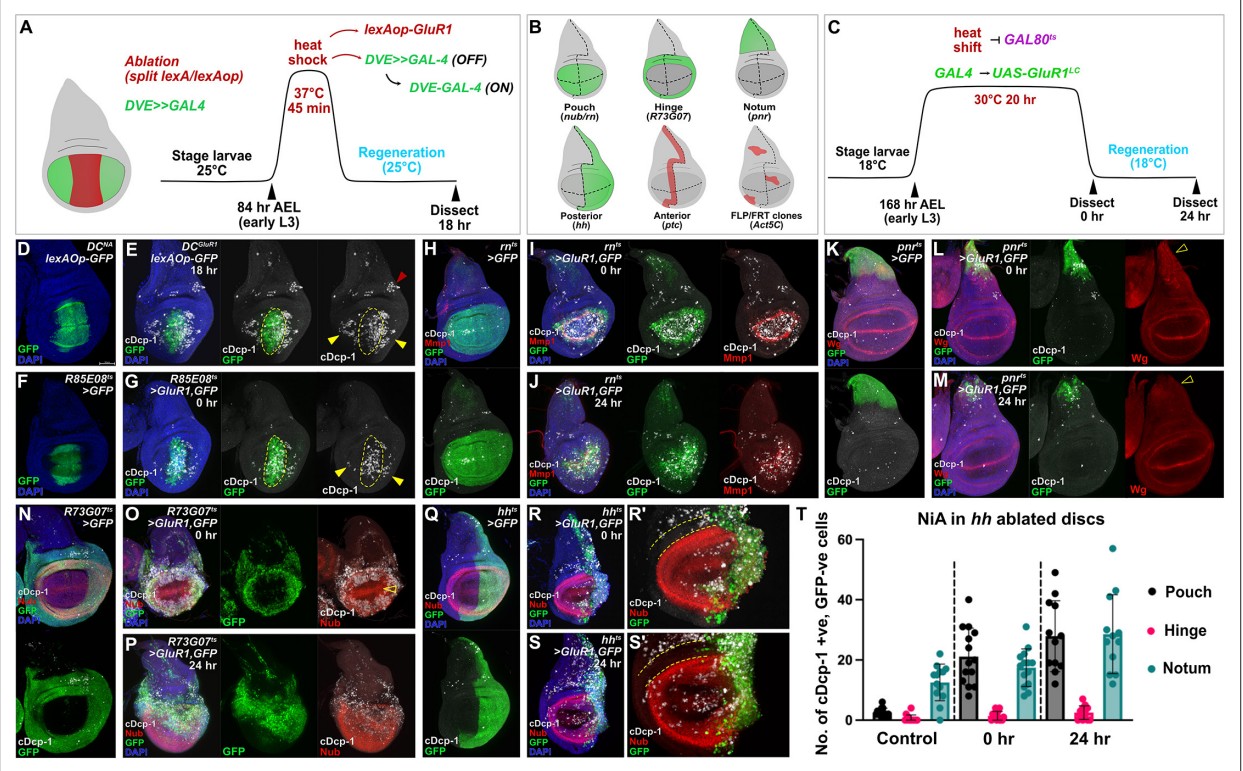

**Figure 1.** The NiA response is restricted to pouch cells following ablation. (**A**) A schematic of the *DC^GluR1* ablation scheme. This system utilizes a split LexA transcriptional activator that promotes the expression of *lexAop-GluR1* following a short heat shock (45 min. at 37 °C). (**B**) A schematic of the different tissue domains and compartments targeted for *GAL4/UAS/GAL80^ts* (*GAL4^ts*) ablation. (**C**) A schematic of the temperature scheme used to induce necrosis with *GAL4^ts>GluR1* following the heat shift (20 hr at 30 °C) induced inactivation of GAL80^ts. (**D**) A control disc for DUAL Control (*DC^NA*) bearing *lexAop-GFP*, which labels the domain targeted for ablation in *DC^GluR1* experiments. (**E**) A *DC^GluR1* ablated disc at 18 hr of regeneration bearing *lexAop-GFP*, yellow arrowheads indicate NiA cells, red arrowhead indicates potential NiA response in the posterior pleura. (**F**) A control disc for *R85E08-GAL4^ts* (*R85^ts>GFP*) ablation. (**G**) An *R85E08^ts>GluR1,GFP* ablated disc, arrowheads indicate the presence of NiA cells following ablation. (**H**) A control disc for whole pouch (*rn^ts>GFP*) ablation. (**I–J**) A *rn^ts>GluR1,GFP* ablated disc dissected immediately following the downshift (0 hr, **I**) and after 24 hr of regeneration (R24, J). (**K**) A control disc for notum ablation (*pnr^ts>GFP*). (**L–M**) *pnr^ts>GluR1,GFP* ablated discs at 0 hr (**L**) and 24 hr (**M**), open arrowheads highlight the absence of the Wg notum stripe following ablation. (**N**) A control disc for hinge ablation (*R73G07^ts>GFP*). (**O–P**) *R73G08^ts>GluR1,GFP* ablated discs at 0 hr (**O**) and 24 hr (**P**). Open arrowhead in (**O**) indicates the absence of NiA in the adjacent wing pouch. (**Q**) A control disc for posterior compartment ablation (*hh^ts>GFP*). (**R-S′**) *hh^ts>GluR1,GFP* ablated discs at 0 hr (**R-R′**) and 24 hr (**S-S′**). Dotted lines in (**R′**) and (**S′**) show an absence of NiA in the anterior hinge. (**T**) A quantification of the NiA response in the anterior pouch, hinge, and notum in control (n=13), 0 hr (n=14), and 24 hr (n=12) discs in response to *hh* ablation. Error bars represent the standard deviation. NiA are defined as cDcp-1-positive (cDcp-1 +ve), GFP-negative (GFP -ve) cells. See **Supplementary file 1** for exact genotypes.

The online version of this article includes the following figure supplement(s) for figure 1:

**Figure supplement 1.** The level of NiA formation is related to the area of necrosis, and can occur in other imaginal discs.

(*rn^ts >GluR1*) or *nubbin-GAL4* (*nub^ts >GluR1*) (**Figure 1H, I and J**, **Figure 1—figure supplement 1A**). Necrosis of the whole pouch with either driver fails to result in NiA (**Figure 1I**, **Figure 1—figure supplement 1A**), even after 24 hr of recovery (**Figure 1J**). Importantly, the majority of cDcp-1-positive cells overlap with expression of the JNK target *Mmp1* (**Figure 1I and J**), and therefore resembles cells undergoing JNK-mediated apoptosis, such as those seen at the wound edge (WE) following distal pouch ablation (**Figure 1—figure supplement 1B**, arrowhead) (**Klemm et al., 2021**). These data suggest that cells outside of the pouch are generally unable to respond to DAMPs released by pouch cells undergoing necrosis to generate NiA, or that such DAMPs are spatially limited.

To test whether necrosis in areas outside of the pouch can induce NiA, we ablated the proximal notum using *pannier-GAL4* (*pnr^ts >GluR1*) (**Figure 1K**), using the absence of the notum Wg stripe to confirm loss of this area (**Figure 1L and M**, open arrowheads). In this ablation, we observed only minimal NiA, with sporadic cDcp-1-positive, GFP-negative cells in the unablated areas of the notum (**Figure 1L**), which does not change after 24 hr (**Figure 1M**). To test the hinge, we used a putative

*zfh1* enhancer driving *GAL4* (*R73G07ts>GluR1*), which has hinge-specific expression (*Figure 1N*). Ablation of the hinge fails to generate NiA in the notum, and surprisingly does not induce a response in the neighboring pouch cells (*Figure 1O and P*), despite their demonstrated ability to form NiA (*Figure 1G*). Given that the regenerative capacity of the wing disc is mainly constrained to the pouch (*Martín et al., 2017*; *Martín and Morata, 2018*), it is possible that the intrinsic differences in regenerative capacity across the disc accounts for the disparity in NiA/NiCP occurrence in the pouch versus the notum. A difference in the availability of the relevant DAMPs or PRRs may also be responsible for the pattern of NiA formation throughout the disc. Alternatively, there may be inhibitory factors present in notum cells that protect against the induction of pro-apoptotic pathway elements. Indeed, differences in caspase 3 activation within the notum and pouch have been shown (*Bergantiños et al., 2010*). Thus, these data suggest that either only the pouch releases DAMPs - and has the requisite PRRs to respond to these DAMPs - that lead to NiA following necrosis, or that unknown genetic factors limit the capacity of the notum and hinge to respond to damage.

As the efficacy of DAMPs might be limited by how far they can reach after being released from lysed cells, we also induced necrosis in the entire posterior disc compartment with *hedgehog-GAL4* (*hhts >GluR1*; *Figure 1Q*), and in an anterior stripe along the anterior/posterior compartment boundary using *patched-GAL4* (*ptcts >GluR1*; *Figure 1—figure supplement 1C*). These experiments cause the simultaneous necrosis of pouch, hinge and notum tissues, allowing us to determine the potential of these different tissue identities to produce NiA. In both experiments, NiA cells are observed in the pouch, and to a lesser extent the notum, but are still strikingly absent from the hinge (*Figure 1R, R' and T*, *Figure 1—figure supplement 1D, D' and F*). After 24 hr of recovery, there is an increase in the number of NiA within the pouch and notum, but not the hinge (*Figure 1S, S' and T*, *Figure 1—figure supplement 1E' and F*). Thus, it appears that NiA can occur outside of the pouch when a large enough area, or an area that also includes the pouch, is ablated. However, the hinge remains completely refractory to NiA formation.

To avoid any bias in the use of tissue-specific *GAL4* drivers, we also made RFP-labeled stochastic clones that have the potential to undergo necrosis upon changing the growth temperature to 30 °C (*Figure 1—figure supplement 1G*). After allowing these clones to develop, we triggered necrosis in early third larval instar and examined the extent of active caspase in the different disc regions (*Figure 1—figure supplement 1H*). As expected, necrosis of clones in the pouch leads to active caspases both within and surrounding the ablated area, including cells without the RFP clone label, suggesting that NiA has occurred (*Figure 1—figure supplement 1H'*, arrowheads). We also found that necrosis in the notum leads to comparatively little caspase labeling (*Figure 1—figure supplement 1H''*) consistent with the notum being less able to generate NiA. Necrotic clones in the hinge also produces caspase activity, but most of these cells also have RFP, suggesting again that NiA does not occur in the hinge (*Figure 1—figure supplement 1H'*). Using this approach, we also examined whether NiA formation occurs in other larval tissues as well, and found that necrotic clones in both haltere and leg discs appear to result in an NiA response (*Figure 1—figure supplement 1J*). Of note, the use of clones that naturally vary in size also demonstrates that the area of ablation is related to the amount of NiA produced (*Figure 1—figure supplement 1I*), which is also true of our tissue-specific ablation experiments (*Figure 1—figure supplement 1I*).

Together, these data infer three important conclusions: (1) all areas of the disc can be killed by necrosis and therefore potentially can release DAMPs, (2) NiA is limited to the pouch when local necrosis occurs, but when multiple (or large) areas of the disc are killed, limited NiA can also be induced in the notum, although we cannot rule out that this is due to DAMPs from dying pouch cells, and (3) the hinge is refractory to NiA, which is consistent with other findings that show its resistance to apoptosis in response to irradiation (*Verghese and Su, 2016*). Thus, the overall pattern of competence to undergo NiA appears to reflect the uneven regenerative capacity of the wing disc, with NiA formation predominantly associated with the highly regenerative wing pouch.

## NiA is regulated by WNT and JAK/STAT signaling

As NiA readily occurs in the pouch but is excluded from the nearby hinge, we used this contrasting response to identify genetic factors that might regulate NiA formation. The wing hinge is specified by JAK/STAT signaling during disc development, which can protect cells from irradiation-induced apoptosis potentially via the expression of *Zn finger homeodomain 2* (*zfh2*; *La Fortezza et al., 2016*;

*Verghese and Su, 2016*; *Verghese and Su, 2018*). Alongside JAK/STAT, the presence of Wingless (Wg, *Drosophila* Wnt1), which encircles the pouch, may also protect cells from death and permit regeneration of the pouch through the repression of *reaper* (*rpr*; *Verghese and Su, 2016*). As such, we investigated both JAK/STAT and Wg to determine whether they regulate NiA formation.

The activity of the JAK/STAT pathway can be visualized in the hinge of early third instar larval discs by a *10XSTAT-GFP* reporter (*Bach et al., 2007*; *Figure 2A*). Upon ablation with $DC^{GluR1}$, high levels of JAK/STAT activity are observed in the pouch at the immediate WE (*Figure 2B*, arrowhead), similar to its upregulation following irradiation or apoptotic ablation (*Herrera and Bach, 2019*). As JNK signaling is induced at the WE (*Figure 1—figure supplement 1B*; *Klemm et al., 2021*), and the *unpaired* ligands are targets of JNK signaling (*Katsuyama et al., 2015*; *Jaiswal et al., 2023*), this JAK/STAT activity is likely to be JNK-mediated. By contrast, low levels of JAK/STAT activity are observed in the areas of the pouch where NiA occurs (*Figure 2B*, open arrowheads), surrounded by the higher developmental JAK/STAT in the hinge (*Figure 2B*). To determine if low JAK/STAT activity is important for NiA, we knocked down the receptor *domeless* (*UAS-dome^RNAi*) in the pouch, which results in a significant increase in NiA (*Figure 2C, D and E*), but has no effect when expressed without damage (*Figure 2—figure supplement 1A*), suggesting that JAK/STAT signaling may negatively regulate the formation of NiA. To support this finding, we next ectopically activated JAK/STAT (*UAS-hop48A*). However, even in the absence of damage, this expression results in high levels of caspase positive cells (*Figure 2—figure supplement 1B* and C), making it difficult to determine an effect on NiA formation. Therefore, to further investigate if JAK/STAT regulates NiA formation, we asked whether reducing developmental JAK/STAT in the hinge might lead to NiA spreading further into this region. We generated a version of DUAL Control that expresses *GAL4* in the posterior compartment by replacing the pouch-specific *DVE >>GAL4* with *hh-GAL4* (*Figure 2F and G*, *Figure 2—figure supplement 1D*). To prevent *hh-GAL4* from being active throughout development, we included $GAL80^{ts}$ (hereafter $DC^{GluR1}hh^{ts}$) and used temperature changes to limit GAL4 activity to the period just prior to ablation (*Figure 2F*). With this system, we knocked down the expression of the JAK/STAT transcription factor *Stat92E* (*UAS-Stat92E^RNAi*) in the posterior compartment (*Figure 2—figure supplement 1E and F*) and ablated the distal pouch, which again shows an increase in caspase-positive cells in the pouch (*Figure 2H*, arrowhead, quantified in *Figure 2L*), but surprisingly NiA cells are still not observed in the hinge (*Figure 2H*, open arrowhead). To further test this result, we also targeted *zinc finger homeodomain 2* (*Zfh2*), a downstream target of the JAK/STAT pathway that potentially protects cells from apoptosis (*La Fortezza et al., 2016*; *Verghese and Su, 2018*). The knockdown of *Zfh2* slightly increases the formation of NiA, although not significantly (*Figure 2—figure supplement 1I and J*, quantified in *Figure 2L*). However, we noted that the level of Zfh2 protein did not appear strongly affected by this knockdown (*Figure 2—figure supplement 1I*), and therefore we cannot confidently conclude that Zfh2 is involved in caspase activation in the context of necrotic damage. Expression of these RNAi lines targeting *Stat92E* or *Zfh2* under non-ablating conditions does not yield any increase in caspase signal (*Figure 2—figure supplement 1G and I*). Thus, JAK/STAT signaling appears to limit NiA formation in the pouch, while the inability for NiA to expand into the hinge upon reducing JAK/STAT suggests that other hinge-specific factors may be involved.

Wg has also been shown to protect cells from apoptosis in the hinge (*Verghese and Su, 2016*), and therefore could influence the formation of NiA. Unlike the stochastic temperature change-induced apoptosis (*Figure 2I*), we noted that NiA in the pouch frequently occurs in discrete populations that avoid Wg at the margin stripe and the inner Wg circle at the boundary of the pouch and hinge (*Figure 2J*), limiting formation of the NiA cells to regions that appear to overlap the *vestigial quadrant enhancer* (*vgQE-lacZ*, *Figure 2—figure supplement 1K and L*; *Kim et al., 1996*). By contrast, cDcp-1-positive cells at the WE do not avoid the Wg margin stripe (*Figure 2J*, arrowhead), suggesting that this behavior may be specific to NiA. To test this, we utilized $DC^{GluR1}hh^{ts}$ to knock down *wg* in the posterior compartment of the disc (*UAS-wg^RNAi*) and found that NiA now occurs in areas of the pouch where *wg* expression is lost, including the wing margin and inner hinge (*Figure 2K*, arrowhead, quantified in *Figure 2L*), unlike NiA in the anterior (*Figure 2K*, open arrowhead). The increase in cDcp-1-positive cells does not occur when *wg* signaling is similarly blocked without damage (*Figure 2—figure supplement 1H*). Notably, the converse experiment in which *wg* is ectopically expressed during ablation does not suppress NiA (*Figure 2—figure supplement 1M and N*), consistent with our hypothesis that other factors, such as targets downstream of JAK/STAT, might act alongside Wg to regulate NiA.

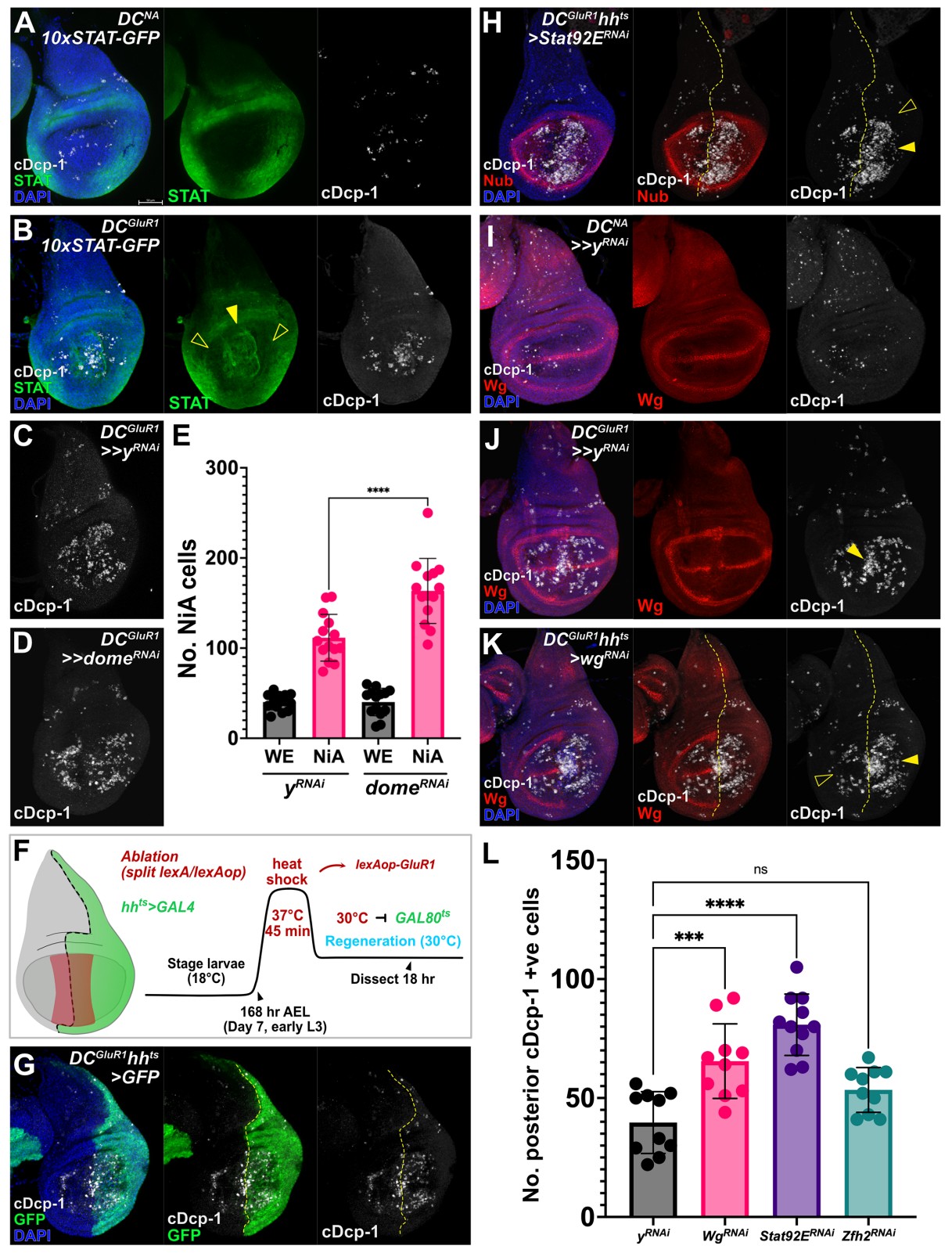

**Figure 2.** NiA formation in pouch cells is regulated by the Wg and JAK/STAT pathways. (**A**) A control disc (*DC^NA^*) bearing the *10xSTAT-GFP* reporter, showing the hinge-specific Stat92E activity that is normally absent from the wing pouch. (**B**) A *DC^GluR1^* ablated disc bearing *10xSTAT-GFP*. Following damage, pouch-specific reporter expression is observed at high levels at the wound edge (arrowhead) and low levels at the NiA area of the pouch (open arrowheads). (**C–E**) *DC^GluR1^>>y^RNAi^* (**C**, n=14) versus *DC^GluR1^>>dome^RNAi^* (**D**), n=14 ablated discs, with the number of NiA quantified in (**E**), P****<0.0001.

*Figure 2 continued on next page*

*Figure 2 continued*

Data were analyzed with a one-way ANOVA followed by a multiple comparisons test. Error bars represent the standard deviation. (**F**) A *DC^{GluR1}hh*-*GAL4* ablation schematic (*DC^{GluR1}hh^{ts}*). A short heat shock (45 min at 37 °C) induces split *LexA/lexAop-GluR1* ablation, while a heat shift (20 hr at 30 °C) following the heat shock will inactivate GAL80^{ts} and permit *UAS*-transgene expression. (**G**) A *DC^{GluR1}hh^{ts}>GFP* ablated disc, where GFP highlights the area of the disc being targeted for JAK/STAT and Wg knockdown in the following experiments. The posterior NiA will be assayed in response to knockdown while anterior NiA serve as an internal control. (**H**) A *DC^{GluR1}hh^{ts}>Stat92E^{RNAi}* ablated disc, arrowhead highlights an increase in pouch NiA, open arrowhead indicates the lack of cDcp-1 signal in the hinge. (**I**) A control *DC^{NA}* disc showing developmental Wg expression. (**J**) A *DC^{GluR1}>>y^{RNAi}* ablated disc showing that NiA cells avoid the areas of the pouch with high Wg expression, while wound edge apoptotic cells overlap the developmental and damage-specific Wg-expressing cells (arrowhead). (**K**) A *DC^{GluR1}hh^{ts}>wg^{RNAi}* ablated disc with NiA cells observed at the posterior margin (arrowhead) while anterior NiA avoid the Wg margin stripe (open arrowhead). (**L**) A quantification of posterior NiA cells (cDcp-1 +ve cells within the pouch) following a control knockdown (*y^{RNAi}*, n=10), *Stat92E^{RNAi}* (n=10), *wg^{RNAi}* (n=10), and *Zfh-2^{RNAi}* (n=11). p***=0.0002, p****<0.0001. ns = not significant. Data were analyzed with a one-way ANOVA followed by a multiple comparisons test. Error bars represent the standard deviation. See ***Supplementary file 1*** for exact genotypes.

The online version of this article includes the following figure supplement(s) for figure 2:

**Figure supplement 1.** Manipulations of JAK/STAT and Wg signaling that regulate NiA.

Together, these data demonstrate that both WNT and JAK/STAT signaling act to limit NiA, thus potentially constraining it to the pouch following necrosis.

## NiA promotes proliferation late in regeneration

As NiA is spatially regulated by at least two major signaling pathways in the disc, we next focused on how the localization of NiA relates to its role in promoting regeneration. In our previous work, using an E2F reporter (*PCNA-GFP*) we found that the appearance of NiA coincides with localized proliferation in the distal pouch at 18 hr post-ablation close to the wound, which persists at 24 hr post ablation (***Klemm et al., 2021***). However, investigation at subsequent time points of recovery shows that regenerative proliferation continues to increase through 36 hr and 48 hr of regeneration, later than we initially assayed, with a significant increase in E2F reporter expression occurring between 18 and 36 hr in the pouch relative to the whole disc, (***Figure 3A-A'''***, ***Figure 3—figure supplement 1A and B, quantified in C***). To investigate this proliferative response, we used EdU to assay the relative level of cell proliferation in discs throughout regeneration from 18 hr to 48 hr post-ablation with *DC^{GluR1}* (***Figure 3B–B'''***), using folds as landmarks to normalize EdU intensity in the pouch relative to the disc (***Figure 3—figure supplement 1D–D'''***). We also performed the same time course with apoptotic ablation using *DC^{hepCA}* for comparison (***Figure 3C–C'''***). At 18 hr of regeneration following ablation with either *DC^{GluR1}* or *DC^{hepCA}*, cells at the WE have already migrated distally to close the injury, while EdU is absent from an area immediately adjacent to the wound (***Figure 3B and C***). This is consistent with the recently described JNK-mediated pause in proliferation that occurs in regenerating wing discs (***Jaiswal et al., 2023***). At 24 hr, this proliferation-devoid area continues to persist following apoptotic injury (*DC^{hepCA}*) and the EdU signal becomes elevated broadly across the rest of the pouch, representing the formation of a blastema (***Figure 3C'***). By contrast, at 24 hr after necrosis (*DC^{GluRI}*), the EdU label is reestablished in cells around the wound showing that proliferation has restarted in these cells (***Figure 3B'***). Unlike *DC^{hepCA}*, the rest of the pouch does not appear to change its rate of proliferation (***Figure 3B'***). By 36 hr following necrosis, a higher EdU signal occurs broadly across the pouch relative to the rest of the disc, which is significantly stronger compared to discs ablated by apoptosis (***Figure 3B'', C'' and E***). This localization is maintained at 48 hr and remains consistently higher in *DC^{GluR1}* versus *DC^{hepCA}* ablated discs (***Figure 3B''', C''' and E***). Thus, the timeline of recovery from necrosis appears to be distinct from that of apoptotic injury, with the strongest changes in regenerative proliferation occurring at comparatively later stages.

The localized proliferation at 36 hr occurs after the appearance of NiA. We previously showed that blocking the apoptotic pathway by simultaneously knocking down DIAP1 inhibitors *rpr*, *hid* and *grim* (*UAS-mir(RHG)*, ***Siegrist et al., 2010***) throughout the pouch limits the initial change in proliferation at early stages (18–24 hr) and inhibits regeneration (***Klemm et al., 2021***). However, it remains unclear whether this newly observed localized proliferation at 36 hr and 48 hr also relies on a functional apoptotic pathway, and moreover, to what extent this regenerative proliferation relies on the JNK-mediated apoptosis at the WE versus the JNK-independent NiA in the LP. To answer these questions, we blocked apoptosis throughout the pouch and this time examined proliferation in late regeneration

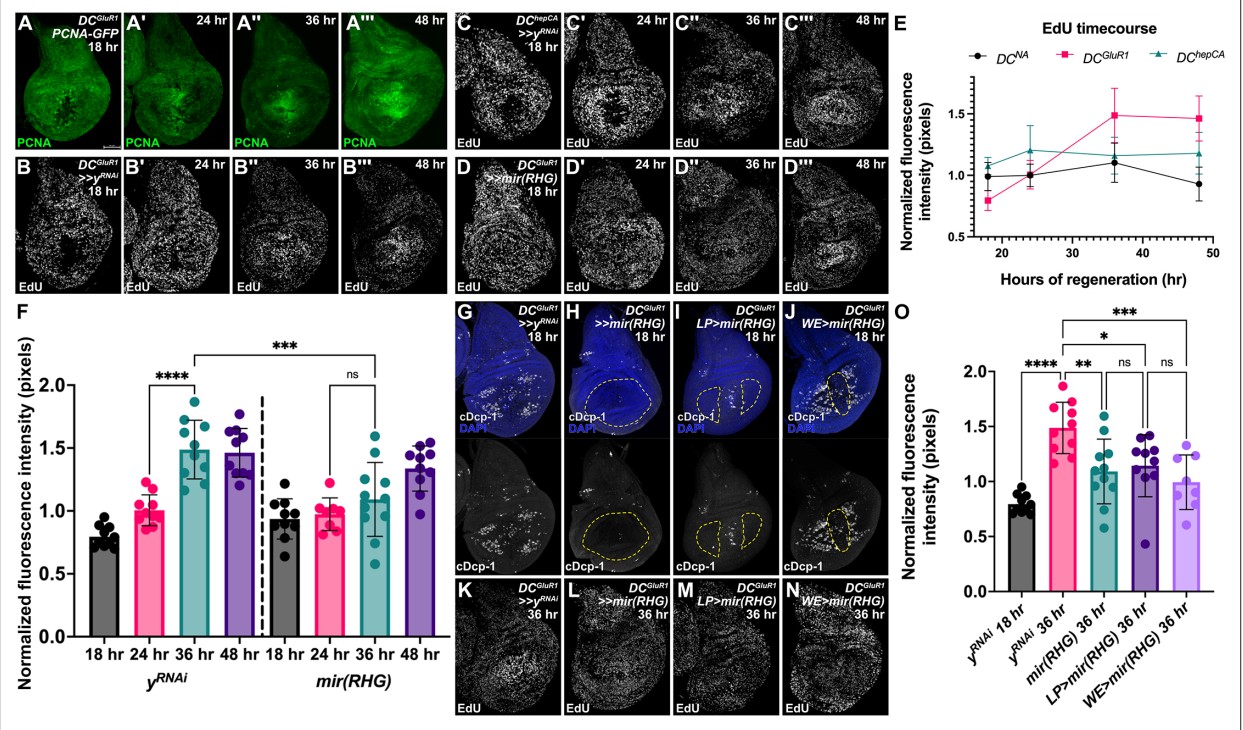

**Figure 3.** NiA promotes proliferation late in regeneration. (A-A''') A time course of $DC^{GluR1}$ ablated discs bearing the *PCNA-GFP* reporter at 18 hr (A), 24 hr (A'), 36 hr (A''), and 48 hr (A'''). (B-B''') A time course of EdU in $DC^{GluR1}$>>$y^{RNAi}$ ablated discs at 18 hr (B), 24 hr (B'), 36 hr (B''), and 48 hr (B'''). Over time, high EdU signal localizes to the damaged pouch by 36 hr and remains elevated at 48 hr. (C-C''') A time course of $DC^{hepCA}$>>$y^{RNAi}$ ablated discs at 18 hr (C), 24 hr (C'), 36 hr (C''), and 48 hr (C'''). (D-D''), demonstrating that EdU initially becomes elevated in the outer regions of the pouch by 24, but then localizes to the central pouch region by 36 hr. A time course of $DC^{GluR1}$>>*mir(RHG)* ablated discs at 18 hr (D), 24 hr (D'), 36 hr (D''), and 48 hr (D'''), demonstrating the loss of EdU localization. (E) A graph of the $DC^{GluR1}$, $DC^{hepCA}$, and $DC^{NA}$ EdU time courses highlight the pattern of EdU labeling in the wing pouch between each system. Error bars represent the standard deviation (F) A quantification of EdU signal intensity between $DC^{GluR1}$>>$y^{RNAi}$ at 18 hr (n=10), 24 hr (n=10), 36 hr (n=10), 48 hr (n=10), and $DC^{GluR1}$>>*mir(RHG)* at 18 hr (n=10), 24 hr (n=8), 36 hr (n=11), and 48 hr (n=10) time courses, ns, not significant; ***p=0.0002; ****p<0.0001; data were analyzed with a one-way ANOVA and multiple comparisons test. Error bars represent the standard deviation. (G–J) $DC^{GluR1}$ ablated discs with different populations of apoptotic cells suppressed by *mir(RHG)*. (G) A control $DC^{GluR1}$>>$y^{RNAi}$ ablated disc showing the typical pattern of NiA formation. (H) A $DC^{GluR1}$>>*mir(RHG)* ablated disc suppressing both wound edge apoptosis and NiA cells. (I) A $DC^{GluR1}$>>*miRHG;DR^{WNT}-GAL80* ablated disc, which targets the NiA area of the pouch for *UAS-mir(RHG)* expression (hereafter Lateral Pouch, or *LP>mir(RHG)*). (J) A $DC^{GluR1}$ x *R85E08>mir(RHG)* ablated disc, which targets wound edge apoptotic cells for suppression (hereafter Wound Edge, or *WE>mir(RHG)*). The dotted lines each panel highlight the area of *UAS-mir(RHG)* expression. (K–N) $DC^{GluR1}$ ablated discs with representative EdU labels at R36 in response to $y^{RNAi}$ (K), whole-pouch *mir(RHG)* (L), *LP>mir(RHG)* (M), and *WE >mir(RHG)* (N). (O) A quantification of the normalized EdU fluorescent intensity of $DC^{GluR1}$>>$y^{RNAi}$ R18 (n=10), $DC^{GluR1}$>>$y^{RNAi}$ R36 (n=10), $DC^{GluR1}$>>*mir(RHG) R36* (n=11), *LP>mir(RHG)* (n=10), and *WE>mir(RHG)* (n=9) ablated discs; *p=0.0189, **p=0.0044, ***p=0.0008, ****p<0.0001; data were analyzed with a one-way ANOVA and multiple comparisons tests. Error bars represent the standard deviation. LP = lateral pouch, WE = wound edge. See *Supplementary file 1* for exact genotypes.

The online version of this article includes the following figure supplement(s) for figure 3:

**Figure supplement 1.** Additional quantification of proliferation in ablated discs and unblated discs.

using EdU (*Figure 3D–D'''*). We found that the localized increase in EdU signaling at 36 hr is lost (*Figure 3D'' and F*), although by 48 hr this increase is mostly restored (*Figure 3D''' and F*). Importantly, the expression of *mir(RHG)* does not influence EdU levels in the absence of damage (*Figure 3— figure supplement 1E, F and G*). Together, these data confirm that a functional apoptotic pathway is necessary to induce a localized proliferative response late in regeneration following necrosis. To understand the relative contribution of WE apoptosis or the NiA in the LP, we designed experiments to block apoptosis in each disc area alone (*LP>mir(RHG)*, *Figure 3I*, and *WE>mir(RHG)*, *Figure 3J*, see Materials and methods for genotypes) relative to the whole pouch knock down (*Figure 3G and H*). Strikingly, the high levels of EdU normally present at 36 hr are strongly reduced when apoptosis is blocked in either population (*Figure 3K–O*), suggesting that both dying cells at the WE and the NiA in the LP contribute to regenerative proliferation following necrosis. These data agree with our

previous findings that both populations are necessary for the overall ability to regenerate adult wings (*Klemm et al., 2021*).

## NiA does not promote proliferation through AiP

The question remains as to how NiA promotes regenerative proliferation. In *Drosophila*, cells undergoing apoptosis secrete factors such as Wg and Dpp to induce the proliferation of neighboring cells as part of a JNK-dependent AiP (*Fogarty and Bergmann, 2017*). Although NiA occurs independent of JNK, to determine whether NiA-induced proliferation relies on any of the same signaling factors as AiP, we examined the damage-specific expression of these various secreted factors. To ensure we could visualize such signals, we used *lacZ*-based reporters and generated undead cells by expressing the baculoviral P35 (*UAS-P35*) in the whole pouch. This protein inhibits activity of the effector caspases Drice and cDcp-1 to block cell death (*Hawkins et al., 2000*; *Meier et al., 2000*), thus allowing signals produced by these cells to be readily detected. Following ablation with $DC^{GluR1}$, ectopic *wg* and *dpp* expression (*wg-lacZ* and *dpp-lacZ*) is observed at the WE (*Figure 4C and C' versus A-B'*, and *Figure 4F and F' versus D-E', arrowheads in C' and F'*) coinciding with JNK activity in this region (*Klemm et al., 2021*). However, *lacZ* is not observed in the LP where NiA occurs (*Figure 4C' and F'*, open arrowheads), indicating that these cells do not activate these mitogens. Similarly, we did not see expression of the EGF ligand *spitz* (*spi-lacZ*) in $DC^{GluR1}$ ablated discs (*Figure 4—figure supplement 1A and B*), which is observed during AiP in the eye (*Fan et al., 2014*). These results suggest that NiA does not promote proliferation through the same signaling factors as those seen during AiP.

We also tested whether other elements required for AiP are involved in NiA-induced proliferation. In addition to mitogen production, AiP also involves the production of extracellular reactive oxygen species (ROS) through a non-apoptotic function of Dronc that activates Duox. (*Fogarty et al., 2016*; *Fogarty and Bergmann, 2017*; *Amcheslavsky et al., 2018*; *Diwanji and Bergmann, 2018*). We first examined the extent of ROS production using dihydroethidium (DHE). This assay showed high levels of ROS localized to the WE but not in the LP (*Figure 4G*, arrowhead), suggesting that NiA does not produce ROS during regeneration. Consistent with this finding, the removal of ROS through pouch-wide expression of either the ROS chelators *Catalase* and *Superoxide dismutase 1* (*UAS-Cat; UAS-Sod1*) or knockdown of *Duox* (*UAS-Duox^RNAi*) has no observable effect on the appearance of NiA (*Figure 4H and I*), although apoptosis at the WE is strongly suppressed in both experiments (*Figure 4H and I*, open arrowhead). It has also been shown that the Duox maturation factor *moladietz* (*mol*) is upregulated following injury to sustain the production of ROS (*Khan et al., 2017*; *Pinal et al., 2018*). However, while a minor increase in lacZ expression occurs at the WE (*Figure 4—figure supplement 1C and D*, arrowhead), no change in the expression of a *mol* reporter (*mol-lacZ*) is observed in response to necrosis in the LP. Finally, to functionally test whether AiP is required for the proliferation associated with NiA, we examined EdU levels across the disc at 36 hr when P35 is expressed. Normally, when undead cells are created via P35, ectopic mitogen production results increased proliferation and tumorous overgrowth. However, when P35 is expressed solely in the LP we saw no change in EdU labeling versus controls (*Figure 4J, L and M*), suggesting NiA do not form undead cells. When expressed in the whole pouch, P35 has a small but non-significant effect on EdU (*Figure 4J, K and M*), consistent with undead cells now being generated at the WE. Together, these data indicate that cells at the WE undergo AiP to contribute to regenerative proliferation, while NiA promote proliferation through a different mechanism.

## A subset of cells undergoing NiA are both caspase-positive and have markers of DNA repair and proliferation

Robust populations of NiA appear in the wing pouch at around 18 hr of regeneration, (*Klemm et al., 2021*), while the localized change in EdU labeling that encompasses much of the damaged pouch is detected later in regeneration at 36 hr and 48 hr (*Figure 3B–B'''', E and F*). Since the loss of NiA abolishes this change in proliferation (*Figure 3F, L–M and O*), we sought to understand how NiA might be influencing proliferation at these later time points. To do so, we used a robust sensor for the activity of the effector caspases cDcp-1 and Drice called *Green Caspase-3 Activity indicator* (*UAS-GC3Ai*, *Schott et al., 2017*) to label NiA throughout regeneration (*Figure 5A–E*). GC3Ai consists of a cyclized GFP with the N and C termini linked by a short peptide that contains a caspase-3 cleavage site (DEVD), which prevents fluorescence under non-apoptotic conditions. Activated effector caspases can cleave

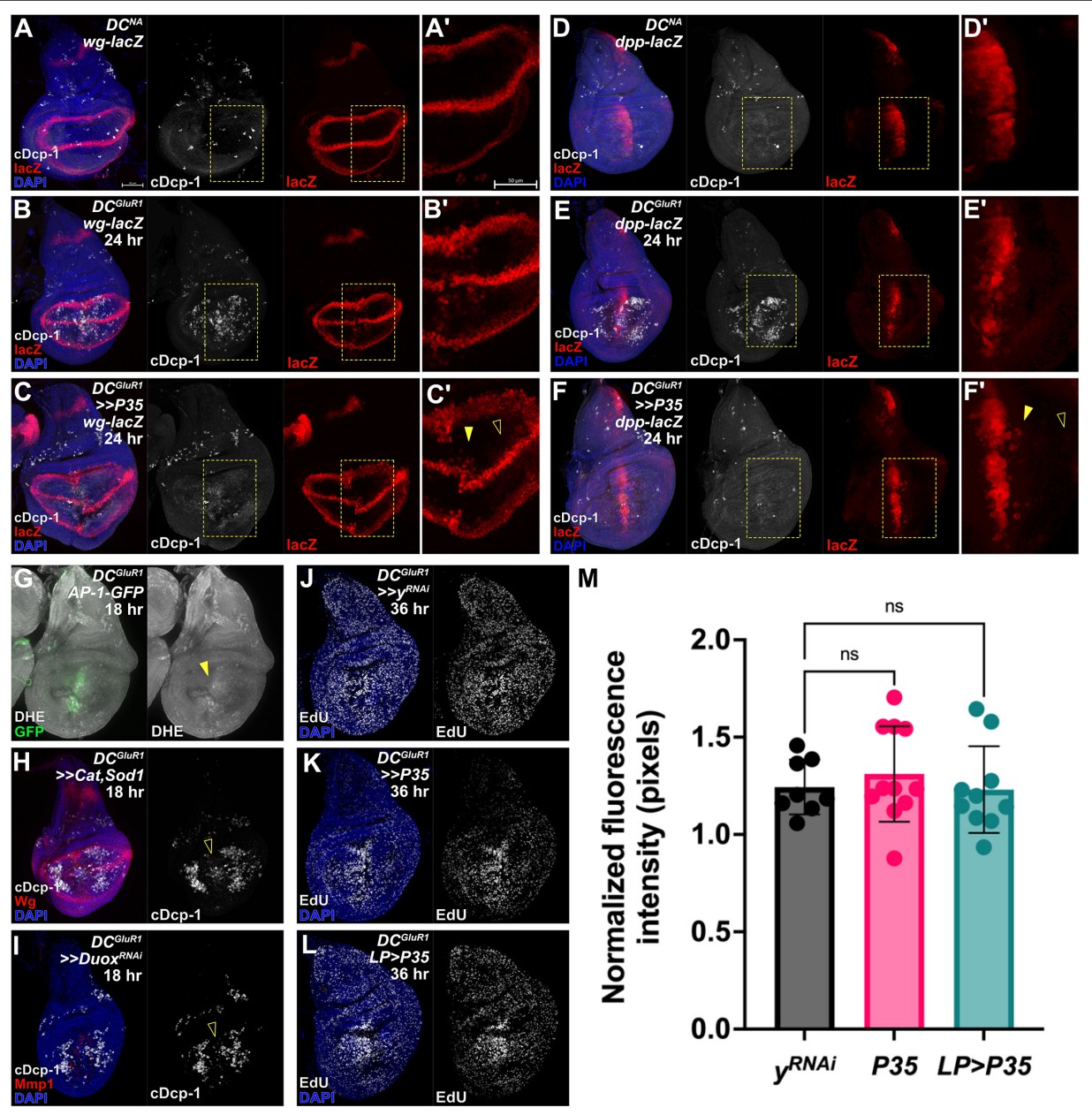

**Figure 4.** NiA do not secrete mitogens associated with Apoptosis-induced Proliferation (AiP). (**A-A'**) A $DC^{NA}$ control disc bearing the *wg-lacZ* reporter. (**B-B'**) A $DC^{GluR1}$ ablated disc with *wg-lacZ* showing the expression of the reporter during regeneration, note several *lacZ*-expressing cells at the wound edge. The 24 hr time point was chosen as an intermediate between NiA formation (18 hr) and blastema formation (36 hr), as any secreted factors involved in AiP are likely to be detected prior to 36 hr. (**C-C'**) A $DC^{GluR1}>>P35$ ablated disc bearing *wg-lacZ* with *lacZ* expressing cells observed at the wound edge (arrowhead) but not in the lateral pouch where NiA occur (open arrowhead). (**D-D'**) A $DC^{NA}$ control disc bearing the *dpp-lacZ* reporter. (**E-E'**) A $DC^{GluR1}$ ablated disc with *dpp-lacZ*; *lacZ*-expressing cells are again observed at the wound edge. (**F-F'**) A $DC^{GluR1}>>P35$ ablated disc bearing *dpp-lacZ*, with *lacZ*-expressing cells at the wound edge (arrowhead) but not where NiA cells occur (open arrowhead), demonstrating that NiA do not secrete factors (Dpp and Wg) involved in AiP. (**G**) A $DC^{GluR1}$ ablated disc bearing the *AP-1-GFP* reporter and labeled with dihydroethidium (DHE) to visualize reactive oxygen species (ROS). Low levels of DHE labeling are observed at the wound edge overlapping *AP-1-GFP* expression (arrowhead), while no DHE labeling is observed in the lateral pouch where NiA occur. (**H**) A $DC^{GluR1}>>Cat,Sod1$ ablated disc showing reduced wound edge apoptosis (open arrowhead) but no change in NiA formation. (**I**) A $DC^{GluR1}>>Duox^{RNAi}$ ablated disc showing a similar pattern to (**H**) with an observed loss of wound edge apoptosis (open arrowhead) but no change in NiA formation. (**J–L**) $DC^{GluR1}$ ablated discs bearing *yRNAi* (**J**), *P35* (**K**), and *LP>P35* (**L**) at 36 hr and labeled with EdU. (**M**) A quantification of discs in (**J–L**) demonstrate no change in EdU labeling upon the expression of P35 in the whole pouch (**K**) or in NiA cells (*LP>P35*, **L**), confirming that NiA promote proliferation independent of AiP, (*y^{RNAi}*, n=8, *P35*, n=11, *LP >P35*, n=10), ns, not significant. Data were analyzed

*Figure 4 continued on next page*

Figure 4 continued

with a one-way ANOVA followed by a multiple comparisons test. LP = lateral pouch, AiP = apoptosis-induced proliferation. See *Supplementary file 1* for exact genotypes.

The online version of this article includes the following figure supplement(s) for figure 4:

**Figure supplement 1.** Changes in *spi* and *mol* expression with ablation.

the linker peptide, resulting in a fluorescent GFP signal that reports caspase activity in real time. Schott et al. demonstrated that apoptotic cells labeled by GC3Ai are extruded within 1 hr of labeling, consistent with previously published work on apoptotic cell extrusion in imaginal discs (*Monier et al., 2015*). With this reporter we found that, unlike normal apoptotic cells that are rapidly cleared from the wing disc, NiA persist and increase in abundance by 36 hr when proliferation localizes to the distal pouch (*Figure 5C*, quantified in *Figure 5F*), as well as at 48 hr (*Figure 5D*, quantified in *Figure 5F*), and even up to 64 hr (wandering stage), when regeneration is complete and pouch tissue is restored (*Figure 5E*, arrowhead). These persistent *GC3Ai*-positive cells are also frequently associated with adjacent cells actively undergoing mitosis by 36 hr, indicated by PH3 labelling (*Figure 5G–G'*). Since the position of NiA cells change over time from 18 hr when they first appear to 36 hr when the change in proliferation occurs, to confirm that these GC3Ai-positive cells originally derive from NiA in the LP rather than the WE, we restricted *GC3Ai* expression to this part of the pouch (*LP>GC3* Ai, *Figure 5H and I*). The *GC3Ai* construct is tagged with an HA epitope that can be used to show its expression even in the absence of activation by caspases (*Figure 5—figure supplement 1A and B*), which we used to confirm that its expression is limited to the LP (*Figure 5H and I*, *Figure 5—figure supplement 1C*). With this experimental setup, we found that *GC3Ai*-positive cells are still present throughout the pouch at 36 hr, including the distal *salm* domain (*Figure 5I*, arrowhead, versus 5 H, open arrowhead, quantified in *Figure 5—figure supplement 1D*), confirming that the source of persistent caspase-positive cells is indeed NiA rather than the WE.

As NiA cells appear to be maintained in the disc for an extended period, we wondered how their behavior and morphology compared to that of normal apoptotic cells. When apoptosis is induced in wing discs using $DC^{hepCA}$, GC3Ai labeling shows apoptotic cells present in the pouch at 18 hr (*Figure 5J*) throughout the disc proper (*Figure 5J'*). At 36 hr these cells appear pyknotic and are basally extruded by 64 hr (*Figure 5K and K'*). By comparison, upon ablation with $DC^{GluR1}$, caspase-positive cells are seen to occupy different regions of the disc at 18 hr, with WE apoptotic cells closer towards the basal surface and NiA derived from the LP still within the disc proper (*Figure 5L and L'*). By 36 hr, the NiA form two distinct populations, some that are rounded up and appear closer to the basal surface, similar to the WE cells (*Figure 5M and M'* red arrowhead), and others that continue to exhibit a columnar appearance and contact both the apical and basal surfaces of the disc (*Figure 5M and M'*, yellow arrowheads). The appearance and position of these cells suggest that a proportion of NiA cells may fail to complete apoptosis but instead persist into late stages of regeneration, despite the presence of detectible caspase activity. This is supported by the observation that only a minority of GC3Ai-positive cells have blebbing and pyknotic nuclei (*Figure 5G'*, white arrowhead), while the majority appear to have a consistent and undisturbed cytoplasmic fluorescent label (*Figure 5G'*, yellow arrowhead). To test our hypothesis that these cells are not undergoing apoptosis, we performed a TUNEL assay to fluorescently label cells with double stranded DNA breaks in GC3Ai-expressing discs. We found that most GC3Ai-labeled cells are co-labeled by TUNEL in early (18 hr) and late (36 hr) stages of regeneration (*Figure 5—figure supplement 1E* and F, arrowheads). However, while TUNEL is associated with apoptosis, by itself it does not confirm that cells are dying (*Grasl-Kraupp et al., 1995*). Therefore, we also examined levels of γH2Av, a histone variant associated with DNA repair and inhibition of apoptosis following damaging stimuli such as irradiation (*Madigan et al., 2002*). Interestingly, we found that GC3Ai-positive cells initially have high levels of γH2Av at 18 hr of regeneration, including those at the WE and the NiA (*Figure 5—figure supplement 1G*, arrowheads), which later is lost from the NiA at 36 hr (*Figure 5—figure supplement 1H*, open arrowheads). This indicates that NiA are undergoing active DNA repair rather than apoptosis. Taken together, these data suggest that a majority of NiA may upregulate caspase activity, but rather than undergoing apoptosis, they repair cellular damage and persist in the tissue into late stages of regeneration where they promote proliferation. As such we are referring to this population of persistent and potentially non-apoptotic NiA as Necrosis-induced Caspase Positive (NiCP) cells.

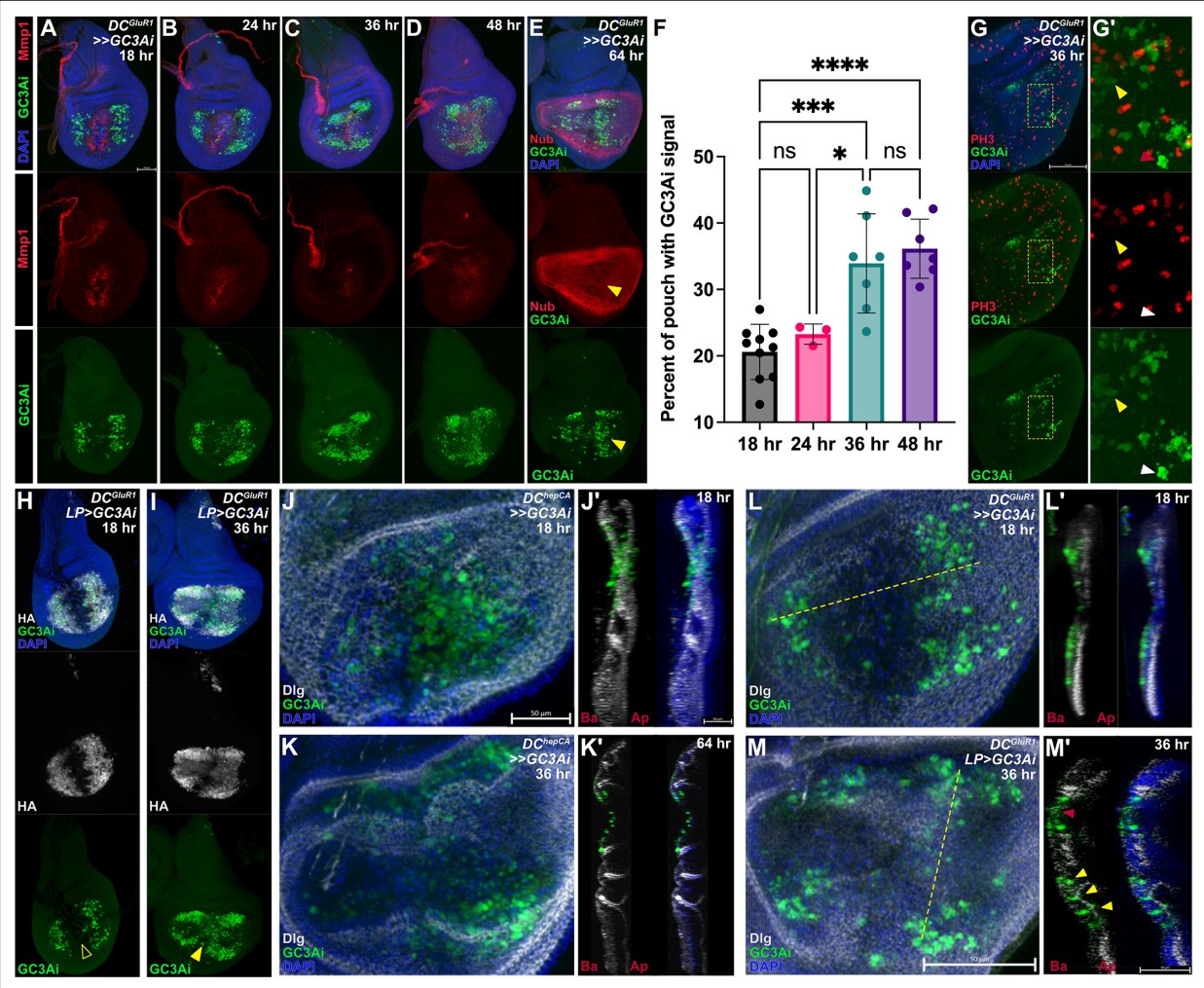

**Figure 5.** NiCP cells persist in the disc proper throughout regeneration. (**A–D**) A time course of *DC^GluR1* ablated discs bearing the fluorescent caspase reporter *UAS-GC3Ai* at 18 hr (**A**), 24 hr (**B**), 36 hr (**C**), and 48 hr (**D**). (**E**) A *DC^GluR1>>GC3*Ai ablated disc at 64 hr, with GC3Ai-positive cells present at the distal pouch after the nubbin marker is reestablished (arrowhead). (**F**) An area quantification of the GC3Ai signal in the whole pouch at 18 hr (n=10), 24 hr (n=3), 36 hr (n=7), and 48 hr (n=7), demonstrating that GC3Ai signal area increases over time. Data were analyzed with a one-way ANOVA followed by a multiple comparisons test. p*=0.0308, p***=0,0001, p****<0.0001. ns = not significant. Error bars represent the standard deviation. (**G-G'**) A *DC^GluR1>>GC3*Ai ablated disc at 36 hr with GC3Ai-positive cells associated with the mitotic PH3 marker. Yellow arrowhead in (**G'**) points to non-apoptotic GC3Ai-positive cells, while the white arrowhead points to GC3Ai labeled cells with an apoptotic morphology. (**H–I**) A *DC^GluR1 LP>GC3*Ai ablated disc at 18 hr (**H**) and 36 hr (**I**). The open arrowhead in (**H**) points to the lack of GC3Ai labeling at the WE. The arrowhead in (**I**) highlights the presence of GC3Ai-labeled NiA cells at the wound edge. (**J-J'**) A *DC^hepCA>>GC3*Ai ablated disc at 18 hr with apoptotic cells present throughout the disc proper. (**K-K'**) A *DC^hepCA >>GC3* Ai ablated disc at 36 hr (**K**) and 64 hr (**K'**); all apoptotic cells appear to be extruded towards the basal surface of the epithelium. (**L-L'**) A *DC^GluR1>>GC3*Ai ablated discs at 18 hr with NiA cells present in the disc proper. (**M-M'**) A *DC^GluR1 LP>GC3*Ai ablated disc at 36 hr, with columnar-shaped GC3Ai-positive cells in the disc proper (yellow arrowhead) alongside pyknotic, rounded GC3Ai-positive cells (red arrowhead), showing that there are a mix of morphologically apoptotic NiA and non-apoptotic NiCP cells in the disc proper. In each of the transverse sections (**J'**, **K'**, **L'**, **M'**), Ba indicates the basal surface of the disc, while Ap indicates the apical surface of the disc. GC3Ai = green caspase-3 activity indicator, LP = lateral pouch. See *Supplementary file 1* for exact genotypes.

The online version of this article includes the following figure supplement(s) for figure 5:

**Figure supplement 1.** Behavior of the *GC3Ai* fluorescent reporter of apoptosis in ablated discs alongside markers of DNA damage and repair.

## NiCP cells have initiator caspase activity but sublethal effector caspase activity

While this characterization of caspase activation and discovery of NiCP cells do not change any of the conclusions we have made up until this point, it does raise the question as to why some cells (NiA) undergo apoptosis and are removed from the tissue in response to necrosis, while others (NiCP) seem

to persist despite caspase activity. The ability for cells to survive caspase activity is not surprising, as many non-apoptotic roles for caspases have been documented (*Su, 2020*), and includes promoting proliferation. As mentioned, the initiator caspase Dronc functions in a non-apoptotic role to activate Duox and thus promote proliferation during AiP in damaged wing discs (*Fogarty et al., 2016*). Therefore, we wondered whether the difference between NiA and NiCP might arise from the level or activity of caspases within these cells. The GC3Ai reporter indicates activity of Dcp-1 and Drice (*Schott et al., 2017*) while the anti-cleaved-Dcp-1 antibody is thought to also detect Drice (*Li et al., 2019*). Thus, these tools exclusively detect effector caspases. As such, to gain a better understanding of caspase activity in NiCP we used two lineage tracing tools to study the activity of initiator versus effector caspases. The initiator caspase-sensitive Drice-Based Sensor (*DBS-GFP*) consists of a modified version of the effector caspase Drice that is (1) a membrane-bound via mCD8, and (2) linked to a nuclear GFP (*Baena-Lopez et al., 2018*). Upon cleavage by Dronc the GFP is released and translocates to the nucleus, providing a readout for Dronc activity. (*Baena-Lopez et al., 2018*). The second tool, CasExpress, is also membrane-bound via mCD8 and attached to GAL4 through a short linker peptide that contains the Drice/Dcp-1 cleavage site DQVD, thus providing a readout for the activity of both effector caspases Drice and Dcp-1 (*Ding et al., 2016*). This tool is highly sensitive and is capable of detecting non-apoptotic caspase activity that occurs during development. However, since GAL4 activity can be modulated by temperature, the sensitivity of CasExpress can be altered by a combination of GAL80$^{ts}$ and temperature to suppress this background signal and optimize damage-induced caspase labeling (*Colon Plaza and Su, 2024*). Our results show a strong overlap of *DBS-GFP* with anti-cDcp-1 in the LP of *DC$^{GluR1}$* ablated discs (*Figure 6A and A'*), indicating that these cells have robust Dronc activity, as do the cells at the WE (*Figure 6A and A'*). However, when we used CasExpress to examine effector caspase activity with a protocol that eliminates developmental caspase activity (*Colon Plaza and Su, 2024*), we noted that only cells at the WE were labelled (*Figure 6B, B' and B''*, arrowhead), most of which had pyknotic nuclei showing they are actively undergoing apoptosis, while few cells in the LP were labelled (*Figure 6B''*). We hypothesized that the level of effector caspase activity might be high enough to be detected by GC3Ai and the cDcp-1 antibody, but not by CasExpress. To test this idea, we further attempted to detect NiCP with CasExpress by combining it with G-TRACE (*Evans et al., 2009*), which should lineage trace cells that have effector caspase activity, permanently labeling them with GFP at the start of ablation. Again, we found that only cells of the WE are labelled during regeneration (*Figure 6B'''*, arrowhead), with minimal labelling of cells in the LP (*Figure 6B''''*, open arrowhead), even after 36 hr of regeneration when the NiCP-induced uptick in proliferation occurs (*Figure 6—figure supplement 1A*). By comparison, performing these same experiments using *DC$^{hepCA}$* to induce extensive apoptotic cell death leads to a significant proportion of cDcp-1-positive cells being labelled by CasExpress under both normal and lineage tracing conditions (*Figure 6C*, arrowhead), suggesting that in the context of necrosis there is not enough effector caspase activity to label NiCP using these methods. Indeed, if the CasExpress experiment is performed in the absence of the GAL80$^{ts}$ that suppresses the background developmental caspase signal, the NiCP cells now become labeled by GFP (*Figure 6D*), indicating that effector caspases are indeed present in these cells, but at low enough levels to avoid death. Evidence for the existence of a cellular execution threshold of caspase activity in cells of the wing disc, which must be reached to induce apoptosis, has previously been documented (*Florentin and Arama, 2012*). This is further supported by the observation that the GFP label becomes more apparent later in regeneration (*Figure 6E*), confirming that cells with effector caspase activity persist rather than die. Thus, it appears that in response to necrosis, the cells of the LP activate the initiator caspase Dronc and the effector caspase(s) Drice/cDcp-1 (to an extent) but fail to undergo programmed cell death. Instead, these cells persist in the disc late into regeneration where they stimulate regenerative proliferation.

## Dronc in NiCP cells promote proliferation independent of AiP

While NiCP cells have both initiator (Dronc) and effector caspase (Drice/cDcp-1) activity, it appears that the level or function of effector caspases is insufficient to cause apoptosis, and is also inconsequential for promoting regeneration – indeed, blocking Drice/cDcp-1 activity with P35 does not affect the localized regenerative proliferation observed at 36 hr (*Figure 4L and M*), or the overall ability to regenerate (*Klemm et al., 2021*). This is in contrast with the ability of effector caspases to drive proliferation in the eye disc (*Fan and Bergmann, 2008a*). However, blocking the apoptotic pathway

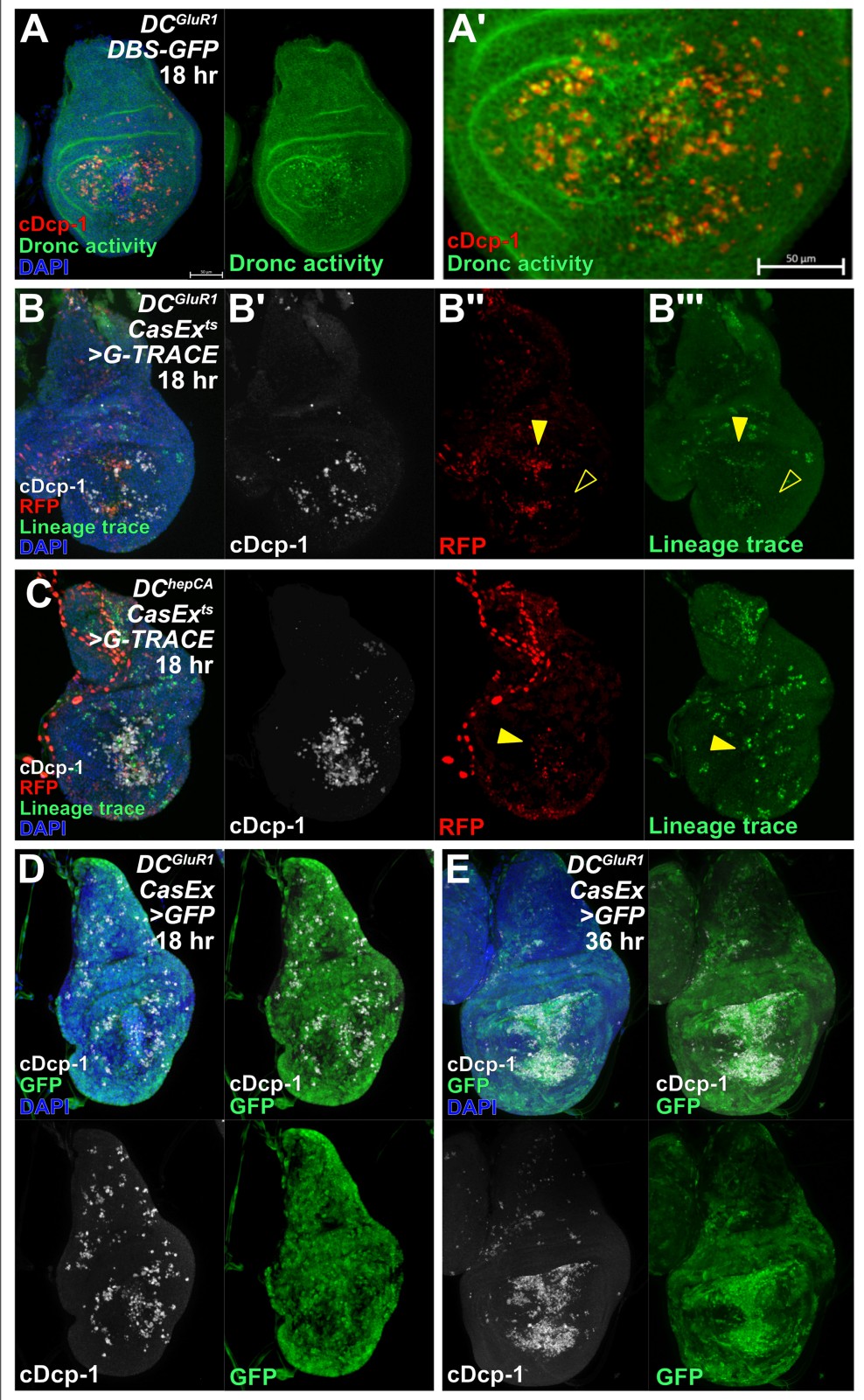

**Figure 6.** NiCP have sublethal levels of activated executioner caspase. (**A-A'**) A *DC^GluR1 DBS-GFP* ablated disc at 18 hr showing strong overlap between cDcp-1 and *DBS-GFP*-positive NiA/NiCP. (**B-B'''**) *DC^GluR1 CasEx^ts>G-TRACE* ablated discs at 18 hr with RFP labeled wound edge cells, (arrowhead), but no RFP or lineage trace-positive labeling is observed in NiA/NiCP (open arrowheads). (**C**) A *DC^hepCA CasEx^ts >G-TRACE* ablated disc at

*Figure 6 continued on next page*

*Figure 6 continued*

18 hr showing a high level of overlap between apoptotic cells and both RFP and the lineage trace (arrowhead). (**D–E**) *DC^GluR1 CasEx>GFP* ablated discs at 18 hr (**D**) and 36 hr (**E**) showing that NiA/NiCP are labeled by *CasExpress* in a sensitized background (when *tubGAL80^ts* is omitted). DBS = Drice-based sensor, CasEx = CasExpress. See ***Supplementary file 1*** for exact genotypes.

The online version of this article includes the following figure supplement(s) for figure 6:

**Figure supplement 1.** Lineage tracing of NiA into late stage regeneration.

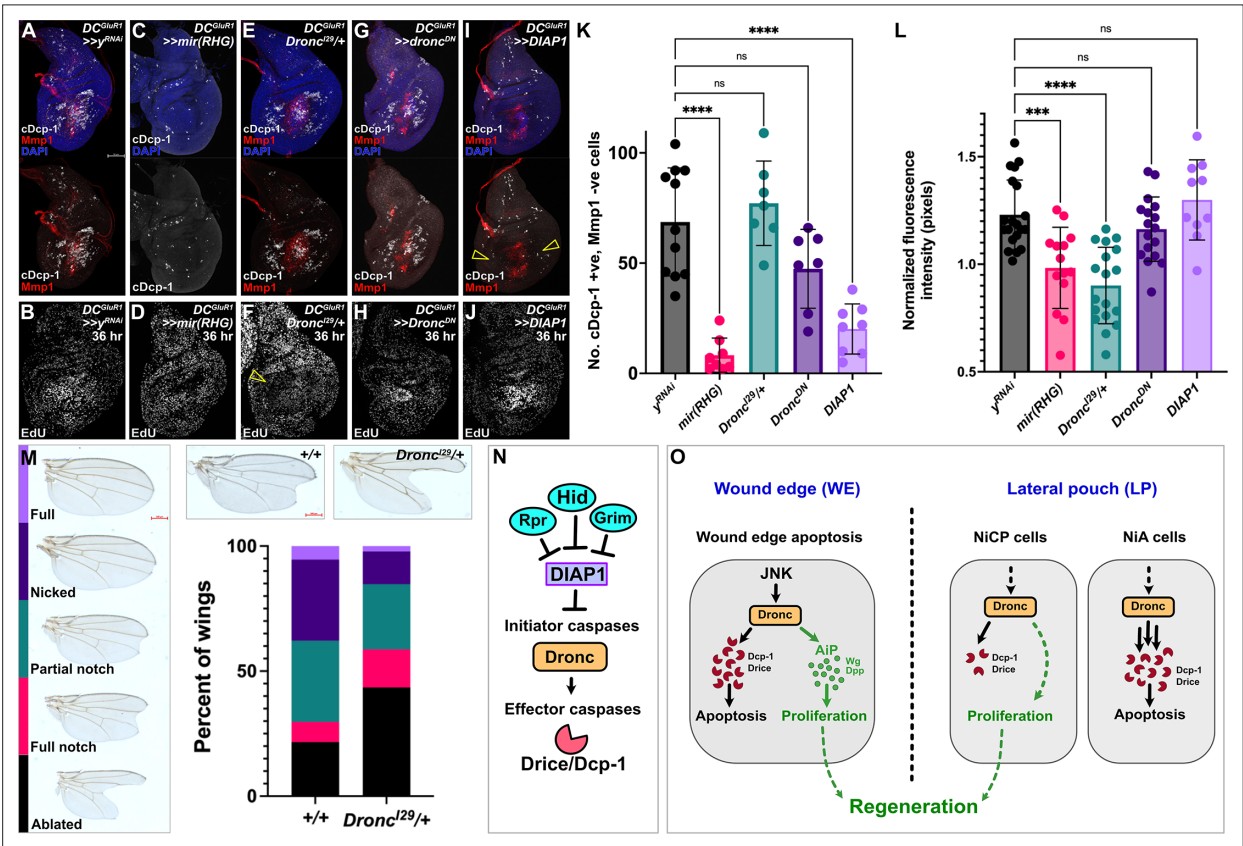

**Figure 7.** Dronc activity in NiCP promotes regeneration. (**A–J**) Representative *DC^GluR1* ablated discs at 18 hr (**A,C,E,G and I**) with cDcp-1 and Mmp1 labeling, and at 36 hr (**B,D,F,H and J**) with EdU labeling, bearing *UAS-y^RNAi* (**A–B**), *UAS-mir(RHG)* (**C–D**), *Dronc^I29/+* (**E–F**), *UAS-Dronc^DN* (**G–H**), and *UAS-DIAP1* (**I–J**). Open arrowhead in (**F**) indicates the strong reduction in EdU labeling. Open arrowheads in (**I**) show a strong suppression of NiA/NiCP as detected by cDcp-1. (**K**) A quantification of NiA/NiCP number (cDcp-1 +ve, Mmp1-ve cells) in *UAS-y^RNAi* (n=11), *UAS-mir(RHG)* (n=8), *Dronc^I29/+* (n=10), *UAS- UAS-Dronc^DN* (n=7), and *UAS-DIAP1* (n=8), ns, not significant, p****<0.0001. Data were analyzed with a one-way ANOVA followed by a multiple comparisons test. Error bars represent the standard deviation. (**L**) A quantification of EdU fluorescence intensity with *UAS-y^RNAi* (n=20), *UAS-mir(RHG)* (n=14), *Dronc^I29/+* (n=18) *UAS- UAS-Dronc^DN* (n=17) and *DIAP1* (n=10), ns, not significant, p***=0.0004, p****<0.0001. Data were analyzed with a one-way ANOVA followed by a multiple comparisons test. Error bars represent the standard deviation. (**M**) Regeneration scoring of *DC^GluR1* ablated flies in a +/+ versus a *Dronc^I29/+* background. Note the increase in frequency of full notched and ablated wings in *Dronc^I29/+* relative to the +/+ treatment. (**N–O**) Schematics depicting the apoptotic pathway in *Drosophila* (**N**) and the three different caspase-positive cell types that occur following necrotic ablation (**O**). Following necrosis, JNK signaling at the wound edge (WE) promotes Dronc activation, which mediates WE apoptosis and proliferation through the AiP feedback loop (left). In the lateral pouch (LP), DAMP-like signal(s) leads to the formation of necrosis-induced caspase positive (NiCP) cells that utilize Dronc to promote proliferation and subsequent regeneration independent of both JNK and AiP signaling, while necrosis-induced apoptosis (NiA) undergo apoptotic death resulting from high levels of dronc, cDcp-1 and Drice activity (right). Both WE apoptosis and NiCP act to promote regeneration through separate mechanisms relying on Dronc-mediated regenerative proliferation. See ***Supplementary file 1*** for exact genotypes.

The online version of this article includes the following figure supplement(s) for figure 7:

**Figure supplement 1.** Quantification of wing size area in a *Dronc* mutant background.

upstream of Dronc using *mir(RHG)* eliminates NiA/NiCP (*Figure 7A, C and K*, *Klemm et al., 2021*), blocks the increase in proliferation (*Figure 7B, D and L*), and limits regeneration (*Klemm et al., 2021*). These observations demonstrate that proliferation induced by NiCP must depend on factors downstream of *rpr/hid/grim* and upstream of the effector caspases *Drice/cDcp-1*, thus potentially pointing to a role for *Dronc*. To test this, we ablated discs with $DC^{GluR1}$ while reducing the activity of Dronc using a null allele ($Dronc^{I29}/+$, *Xu et al., 2005*). With this genetic background there is a significant reduction in regenerative proliferation at 36 hr (*Figure 7F and L*) and regeneration is limited, shown by both adult wing scores (*Figure 7M*) and size (*Figure 7—figure supplement 1A*). These data demonstrate an essential role for Dronc in NiCP to promote proliferation and subsequent regeneration of the disc following necrosis. We noted that this mutant does not strongly affect the number of NiCP assayed by cDcp-1 (*Figure 7E and K*), likely because this allele does not completely suppress apoptosis unless homozygous (*Xu et al., 2005*), which is precluded by the genetics of this experiment. As an alternative, we interfered with Dronc function by expressing a dominant negative form of *Dronc* that contains only the caspase recruitment domain (CARD), here referred to as $Dronc^{DN}$, which blocks activation of Dcp-1/Drice and apoptosis (*Meier et al., 2000*). $Dronc^{DN}$ reduces NiCP number, but to a lesser degree than *mi(RHG)* (*Figure 7G and K*) and does not affect proliferation (*Figure 7H and L*), suggesting that the CARD domain is dispensable for NiCP-induced proliferation.

Finally, we wondered how this might relate to the previously documented role of Dronc in promoting proliferation following apoptotic cell death during AiP (*Fogarty and Bergmann, 2017*). AiP depends on both JNK and ROS (*Fogarty et al., 2016*), which we have shown are only present at the WE and are not associated with NiA/NiCP (*Figure 1—figure supplement 1B*, *Figure 4G, H and I*). Thus, it is possible that Dronc's function in response to necrosis occurs via a distinct mechanism. Importantly, the activity of Dronc in both apoptosis and in AiP is influenced by the Dronc's upstream regulator, DIAP1 (*Meier et al., 2000*). DIAP1 modifies Dronc's CARD domain through K78 mono-ubiquitylation, which blocks both apoptosis (*Kamber Kaya et al., 2017*), and AiP (*Fan and Bergmann, 2008a*; *Fogarty and Bergmann, 2017*). Thus, we expressed DIAP1 (*UAS-DIAP1*) following ablation with $DC^{GluRI}$ and found that it strongly suppresses the number of cDcp-1-positive cells at the WE and the NiA/NiCP in the LP (*Figure 7I and K*), but strikingly has no effect on regenerative proliferation at 36 hr (*Figure 7J and L*). These results demonstrate a key role for the initiator caspase *Dronc* in promoting regenerative proliferation following necrosis, which is not affected by DIAP1, and therefore is likely separate from its role in apoptosis and the AiP mechanism.

Taken together, our data suggest a model in which necrosis leads to the establishment of distinct cell populations important for regeneration (*Figure 7N and O*). After injury, cells at the immediate WE undergo JNK-mediated apoptosis and contribute to proliferation via the established AiP mechanism (*Figure 7O*, left), while cells at a distance from the injury in the LP activate Dronc via an unknown DAMP-like signal(s) that occurs independent of JNK. Some of these cells go on to activate effector caspases at levels high enough to result in apoptosis (NiA, *Figure 7O*, right), while others activate these caspases at a low enough level to be detectable but insufficient to induce death (NiCP, *Figure 7O* right). Instead, these cells persist in the tissue late into the repair process, where they promote proliferation via a novel non-apoptotic and AiP-independent function of the initiator caspase Dronc.

## Discussion

An important and often overlooked factor when studying regeneration is the type of injury – and consequently the type of cell death it causes – that can significantly impact repair processes. The existence of conserved signaling events that promote recovery has been well established in the context of apoptosis (*Fogarty and Bergmann, 2017*; *Pérez-Garijo, 2018*), while the importance of such events in necrosis are less well established. Here, we have investigated the genetic events that occur in the aftermath of necrosis and how they influence the ability of a tissue to recover and regenerate. We previously showed that wing discs can regenerate effectively in the face of necrotic cell death (*Klemm et al., 2021*), triggering local JNK-dependent AiP at the WE, which is likely in response to tissue disruption (*La Marca and Richardson, 2020*), while also inducing caspase activity in cells at a distance from the wound. This induction is independent of JNK signaling and, alongside having effector caspase activity, these cells can be marked by TUNEL and can be blocked by inhibiting the apoptotic pathway (*Klemm et al., 2021*). As such, we determined that these cells are undergoing PCD and named this phenomenon Necrosis-induced Apoptosis (NiA). We also showed that both NiA

and AiP at the wound contribute to regeneration. Our current work further characterizes the NiA phenomenon, and we have now shown that cells undergoing NiA actually comprise two populations with separate behaviors. Upon necrosis, cells of the LP appear to activate effector caspases, indicated by cDcp-1 antibody staining and the transgenic reporter GC3Ai. While a proportion of these cells develop apoptotic morphology, round up and are cleared over time as part of NiA, a large number of cells appear able to persist despite the presence of caspases, where they promote regenerative proliferation dependent on the initiator caspase Dronc (*Figure 7N*). To reflect these new findings, we have called these NiCP cells. While NiCP appear to promote proliferation, it remains unclear whether they do so through cell autonomous or non-autonomous effects. We were able to determine that NiCP do not utilize AiP as a potential mechanism, as these events occur in the absence of JNK, and exhibit none of the established hallmarks of AiP, including the presence and requirement for ROS, the ability to be blocked by DIAP1, or the production of mitogens such as Wg and Dpp. However, it is possible that other signaling factors may be secreted by these NiCP cells. Alternatively, the observed non-apoptotic caspase activity in NiCP cells may interact with intracellular factors to promote cell-autonomous proliferation. In either case, our results have identified an important role for Dronc in promoting regeneration in response to necrotic cell death.

## The signal from necrotic cells that leads to NiA/NiCP remains unknown

Although we have shown that necrosis leads to NiCP cells and NiA at a distance from the site of injury, the signal that leads to these events upon wounding is still unknown. Cells undergoing necrosis release DAMPs, a category of molecules that includes both common cellular contents as well as specific proteins, both of which can produce downstream responses like inflammation and the activation of effectors that promote a healing response (*Gordon et al., 2018*). DAMPs of both categories have been demonstrated in *Drosophila*; in apoptotic-deficient larvae, circulating DAMPs in the hemolymph can constitutively activate immune signaling in the fat body (*Nishida et al., 2024*), while specific factors such as α-actinin (*Gordon et al., 2018*), and HMGB1 (*Nishida et al., 2024*) are known to possess DAMP activity. Recent work describing an in vivo sensor for HMGB1 demonstrates the release of this DAMP from wing disc cells in response to a necrotic stimulus similar to that used here (*Nishida et al., 2024*), making this an important candidate to test for a potential role in producing NiCP. However, it is equally possible that the causative DAMP(s) is/are one or more of the common funda-mental cellular components released upon cell lysis, which would be more challenging to manipulate and test. Considering the potential diversity and variable nature of DAMPs, it may be more feasible to instead identify the downstream PRRs that are required to interpret this unknown signal. Several groups of genes can act as PRRs, most notably the Toll-like receptor (TLR) family, various members of which can respond to DAMPs (*Ming et al., 2014*; *Gong et al., 2020*). Nine TLR genes have been iden-tified in *Drosophila*, which have assorted roles in development and innate immunity (*Anthoney et al., 2018*), and have been implicated in DAMP sensing (*Ming et al., 2014*). Thus, it would be valuable to test the requirement for TLRs, alongside other suspected PRRs such as scavenger receptors (*Cao, 2016*; *Roh and Sohn, 2018*; *Gong et al., 2020*) for their requirement to cause NiCP.

An additional approach to elucidating how these cells are generated is by leveraging our finding that both JAK/STAT and WNT signaling seem to block NiCP/NiA in the disc. JAK/STAT signaling promotes survival of cells in response to stress by repressing JNK signaling, thus minimizing JNK-mediated apoptosis (*La Fortezza et al., 2016*). Although NiCP appear to avoid damage-induced JAK/STAT upregulation in the pouch, it is unlikely they are regulated by this mechanism, since NiCP cells occur independent of JNK activity (*Klemm et al., 2021*). However, within the hinge, which is completely devoid of NiCP, JAK/STAT protects cells from apoptosis potentially by upregulating *DIAP1* (*Verghese and Su, 2016*) and *zfh2* (*Verghese and Su, 2018*), while Wg represses the transcription of *rpr*, (*Verghese and Su, 2016*). Both signaling pathways are required autonomously in hinge cells for their ability to replace and regenerate ablated pouch tissue (*Verghese and Su, 2016*; *Ledru et al., 2022*). Our results show that DIAP1 can prevent cDcp-1 activation in NiCP in the pouch, but not the corresponding proliferation that they produce, suggesting one reason that NiCP is not seen in the hinge is possibly due to the high DIAP1 threshold that results from developmental JAK/STAT activity, which blocks Dronc's ability to activate effector caspases in this region. Thus, it remains to be seen whether lowering DIAP1 levels in the hinge, or manipulating Wg to allow *rpr* expression alongside changes in JAK/STAT, could allow NiCP to form in the hinge, and whether this might lead to

regenerative proliferation in this region. Finally, the observation that both NiCP cells and their associated proliferative effects appear to occur independent of JNK signaling, which is normally central in models of stress and damage (*Pinal et al., 2019*), could help to identify the upstream signaling that leads to NiCP following necrosis. Together, these approaches could provide essential information as to the underlying genetic and cellular events that connect the lysis of cells during necrosis to the formation of NiCP cells required for regeneration.

## The persistence of NiCP cells versus their elimination via NiA

Our work shows that cells in the LP can either persist as NiCP and contribute to regenerative proliferation, or progress to apoptosis as part of NiA, but how this decision is made is unknown. During apoptosis in the wing disc, a threshold level of effector caspases must be reached for the cell to complete PCD (*Florentin and Arama, 2012*). Thus, we hypothesize that DAMP signals from necrotic cells may result in inconstant levels of effector caspase activity in cells of the LP – some with high caspases that advance to apoptosis (NiA), recognized by the changes in morphology and position in the disc characteristic of PCD, which are ultimately cleared, and others (the NiCP cells) that have low enough caspase activity levels to survive, but this activity can still be detected by sensitized reagents. This is supported by our observations made using effector caspase-based lineage tracing using CasExpress (*Ding et al., 2016*), in which caspases can only be detected in NiCP cells by lowering the detection threshold. Alternatively, since CasExpress is a membrane-based reporter (*Ding et al., 2016*), it is also possible that effector caspases within NiCP cells have a different subcellular localization that reflects a non-apoptotic function, thus preventing reporter activation, rather than a difference in expression level or activity. The idea that caspases have different functions, both apoptotic and non-apoptotic, based on localization is well established, such as the targeting of specific cleavage substrates in distinct subcellular compartments (*Brown-Suedel and Bouchier-Hayes, 2020*), or the trafficking of Dronc to the membrane to promote ROS (*Amcheslavsky et al., 2018*). Whatever the mechanism is that distinguishes between elimination via NiA or persistence as NiCP, the proportion of LP cells that participate in each remains difficult to assay due to the dynamic nature of cell death (*Nano and Montell, 2024*) and overall variability of the NiCP/NiA phenotype. Anecdotal evidence from our experiments examining GC3Ai-labelled cells over time suggests the majority of cells in fact do not undergo apoptosis and are therefore NiCP. Importantly, however, while our experiments demonstrate that GC3Ai labeled NiCP increase over time, it is unclear whether this increase in signal abundance is due to cells with newly activated caspases that cleave and activate GC3Ai, or if GC3Ai labeled NiCP are instead proliferating. The use of other tools with potentially different sensitivities to effector caspases, such as Apoliner (*Bardet et al., 2008*), or CD8-PARP-Venus (*Florentin and Arama, 2012*), may shed further light on this issue. Nevertheless, it remains to be seen how a cell becomes NiCP or undergoes NiA and dies, and whether distinct levels of caspases (initiator or effector) their localization, or a different attribute is responsible.

## How does NiCP contribute to regeneration?

One of the most important questions that future work must address is how the phenomena we have identified lead to regeneration following necrosis. Our experiments suggest that different caspase-positive populations of cells may contribute to regenerative proliferation: Firstly, the smaller number of apoptotic cells at the WE generated by epithelial disruption likely contribute by the established process of JNK-dependent AiP (*Fogarty and Bergmann, 2017*). Secondly, the caspase-surviving NiCP cells in the LP that contribute via the initiator caspase Dronc. Crucially, this non-apoptotic function of Dronc cannot be inhibited by the expression of DIAP1, which has previously been shown to block both the apoptotic and non-apoptotic AiP functions of Dronc (*Kamber Kaya et al., 2017*). Thus, our results suggest that Dronc acts via a different mechanism to induce growth in response to necrosis. The current understanding of how Dronc functions at the molecular level may provide valuable clues as to how. DIAP1 suppresses Dronc through its E3 ubiquitin ligase activity, which mono-ubiquitylates the caspase recruitment (CARD) domain of Dronc to suppress both apoptotic and non-apoptotic functions related to AiP (*Kamber Kaya et al., 2017*). Here, we find that the ectopic expression of *DIAP1* or just the CARD-containing pro-domain of Dronc (*Dronc^{DN}*) cannot block regenerative proliferation, while by contrast heterozygosity for the *Dronc^{I29}* null allele that can inhibit AiP (*Kamber Kaya et al., 2017*) is sufficient to block NiCP-mediated proliferation. Thus, regulation of Dronc activity in the

context of necrosis may not rely on the modification of its CARD domain. The documented functions of Dronc have also shown a requirement for its catalytic domain; mutations in this domain can block the activation of Drice and AiP-induced overgrowth (*Fan et al., 2014*). Thus, the inability of DIAP1 to suppress NiCP-mediated proliferation may be due to the catalytic activity of Dronc. Importantly, although the CARD and catalytic domains are important points of regulation for Dronc activity, it has also been shown that damage-specific context cues are also vital. For example, the ectopic expression of *Dronc$^{K78R}$*, a mutant that renders it irrepressible by DIAP1, might be expected to induce significant apoptotic cell death. However, this is not the case unless its structural binding partner, the APAF1 ortholog encoded by *Dark*, is expressed alongside (*Shapiro et al., 2008*), demonstrating the importance of stoichiometry between Dronc and Dark for the apoptotic function of Dronc. As such, further investigation into Dronc's functional domains and context-dependent interactions with respect to its CARD and catalytic domains as well as its binding partners, including DIAP1 and Dark, will be necessary to understand how Dronc is involved in promoting regenerative proliferation in response to necrotic injury.

## NiCP as a general mechanism to promote regeneration

Although having initially been characterized for their central role in apoptosis, many non-apoptotic functions of caspases have since been discovered, revealing them to be dynamic regulators of diverse process including cell fate specification, cellular remodeling, tissue growth, development, metabolism and others (*Shinoda et al., 2019*; *Wang and Baker, 2019*; *Su, 2020*). Studies of caspase signaling during regeneration have revealed essential aspects of non-apoptotic caspase activity, such as initiator and effector-dependent models of AiP that contribute to repair (*Ryoo and Bergmann, 2012*; *Fogarty and Bergmann, 2017*). The contrasts in caspase functions that we have observed between apoptotic and necrotic damage, despite ultimately resulting in comparable levels of regeneration in an organ (*Klemm et al., 2021*), underscores the nuance that exists in damage signaling between different injury contexts. It is clear that caspase activity in response to injury as a mechanism to promote regeneration is a highly conserved process that occurs in many organisms, regardless of tissue identify or type of damage incurred (*Bergmann and Steller, 2010*; *Vriz et al., 2014*; *Fuchs and Steller, 2015*; *Pérez-Garijo and Steller, 2015*; *Fogarty and Bergmann, 2017*; *Pérez-Garijo, 2018*). Our findings reinforce the idea that much remains to be understood about the role of caspases in tissue repair, and that the nature of an injury – and therefore the type of cell death involved – is an important aspect that must also be considered.

## Materials and methods
### *Drosophila* stocks

Flies were cultured in conventional dextrose fly media at 25 °C with 12 h light–dark cycles. The recipe for dextrose media contains 9.3 g agar, 32 g yeast, 61 g cornmeal, 129 g dextrose, and 14 g tegosept in 1 L distilled water. Genotypes for each figure panel are listed in the *Supplementary file 1*. Fly lines used as ablation stocks are as follows: hs-FLP; hs-p65; salm-LexADBD, DVE >>GAL4 (DC$^{NA}$), hs-FLP; hs-p65; salm-LexADBD/ TM6C, sb (DC$^{NA}$ no GAL4), hs-FLP; lexAop-GluR1$^{LC}$, hs-p65 /CyO; salm-LexADBD, DVE >>GAL4/TM6 B, Tb (DC$^{GluR1}$), hs-FLP; lexAop-GluR1$^{LC}$, hs-p65/CyO; salm-LexADBD/ TM6C, sb (DC$^{GluR1}$ no GAL4), hs-FLP;lexAOp-GluR1$^{LC}$/ CyO; salm-LexADBD, hh-GAL4 /TM6B, Tb (DC$^{GluR1}$ hh-GAL4), hs-FLP;lexAop-hepCA, hs-p65/CyO; salm-DBD, DVE >>GAL4/TM6B, Tb (DC$^{hepCA}$), UAS-GluR1; tubGAL80$^{ts}$, and tubGAL80$^{ts}$. The stock DR$^{WNT}$-GAL80 was used to limit *UAS*-transgenes to the lateral pouch (LP) where NiA occur (*Klemm et al., 2021*), while *R85E08-GAL4* was used to drive *UAS*-transgene expression at the wound edge (WE). The following stocks were obtained from Bloomington *Drosophila* Stock Center: w$^{1118}$ (BL#3605), UAS-y$^{RNAi}$ (BL#64527), UAS-p35 (BL#5073), UAS-GC3Ai (II, BL#84346), UAS-GC3Ai (III, BL#84343), rn-GAL4 (BL#7405), hh-GAL4 (BL#600186), ptc-GAL4 (BL#2017), pnr-GAL4 (BL#), (BL#25758), nub-GAL4 (BL#25754), R73G07-GAL4(BL#39829), UAS-Zfh2$^{RNAi}$ (BL#50643), UAS-wg$^{RNAi}$ (BL#32994), UAS-Stat92E$^{RNAi}$ (BL#35600), dpp-lacZ (BL#8412), wg-lacZ (BL#50763), spi-lacZ (BL#10462), UAS-p35 (II, BL#5072), UAS-p35 (III, BL#5073), AP-1-GFP (*Chatterjee and Bohmann, 2012*), act >>GAL4, UAS-RFP (BL#30558), DBS-GFP (III, BL#83130), CasExpress (BL#65419), G-TRACE (III, BL#28281), tubGAL80$^{ts}$ (II, BL#7019), tubGAL80$^{ts}$ (III, BL#7017), 10xSTAT-GFP (BL#), UAS-dome$^{RNAi}$ (BL#32860), UAS-hop48A (BL#), PCNA-GFP (BL#25749), UAS-Cat

(BL#24621), UAS-Sod1 (BL#24754), UAS-Duox$^{RNAi}$ (BL#32903), mol-lacZ (BL#12173), UAS-Dronc$^{DN}$ (BL#58992), UAS-DIAP1 (BL#6657), and dronc$^{I29}$/TM3, Sb (BL#98453). UAS-mir(RHG) was gifted from the Hariharan lab at UC Berkeley. vgQE-lacZ was gifted from Tin Tin Su. UAS-GluR1$^{LC}$ (*Liu et al., 2013*) was gifted from the Xie lab at Stowers Institute.

## Ablation experiments

### DUAL control ablation with *DVE>>GAL4*

DUAL control experiments were performed essentially as described in *Harris et al., 2020*. Briefly, experimental crosses were cultured at 25 °C and density controlled at 50 larvae per vial. Larvae were heat shocked on day 3.5 of development (84 hr after egg deposition (AED)) by placing vials in a 37 °C water bath for 45 min, followed by a return to 25 °C. Larvae were allowed to recover for 18 hr before being dissected, fixed and immunolabeled, unless otherwise indicated. *UAS-y$^{RNAi}$* and *UAS*-GFP were used as control lines for RNAi-based experiments. *w$^{1118}$* was used as a control for *dronc$^{I29}$* experiments. The *DVE>>GAL4* driver drives expression in the wing pouch, allowing for the regenerating wound edge cells and NiA cells to be targeted for interrogation (*Klemm et al., 2021*). The *DR$^{WNT}$-GAL80* transgene (*Klemm et al., 2021*) was included as necessary to restrict *UAS*-transgene expression to the lateral pouch (LP), where NiA/NiCP occurs. Ablation experiments were performed in at least three biological repeats (arising from an independent genetic cross). Sample sizes were determined based on phenotype consistency and standards in the field.

### DUAL control ablation without *DVE>>GAL4*

To restrict *UAS-* expression to wound edge apoptotic cells, a version of DUAL Control lacking the DVE>>GAL4 (*DC$^{GluR1}$ no GAL4*) was crossed to the *R85E08-GAL-4* driver. *DC$^{GluR1}$no GAL4* experiments were performed along the same parameters as *DC$^{GluR1}$* experiments. The *R85E08-GAL4* driver was used alongside *DC$^{GluR1}$ no GAL4*, when necessary, to target *UAS*-transgene expression to the wound edge (WE), where the JNK-dependent wound edge apoptotic cells occur.

### DUAL control ablation with *hh-GAL4*

DUAL control flies bearing *hh-GAL4* (*DC$^{GluR1}$hh$^{ts}$*) were cultured at 18 °C and density controlled at 50 larvae per vial. *tubGAL80$^{ts}$* was included to conditionally express *UAS*-based constructs after ablation. Larvae were heat-shocked on day 7 of development (168 hr AED) for 45 min at 37 °C, followed by incubation at 30 °C to inactivate *tubGAL80$^{ts}$* and permit *UAS*-based expression. Larvae were held at 30 °C for 18 hr before being dissected, fixed, and immunolabeled.

### *GAL4/UAS* ablation

*GAL4/UAS*-based ablation experiments were performed essentially as described in *Smith-Bolton et al., 2009*. Briefly, larvae bearing either *UAS-GluR1;tubGAL80$^{ts}$* were cultured at 18 °C and density controlled at 50 larvae per vial. Larvae upshifted on day 7 of development (168 hr AED) for 20 hr at 30 °C and were either immediately dissected (denoted as 0 hr) or were allowed to recover for 24 hr before being dissected, fixed, and imaged. *tubGAL80$^{ts}$* flies were used as a non-ablating control.

### *FLP/*FRT ablation experiments

To generate clonal patches of *UAS-GluR1;UAS-RFP*-expressing cells, flies of the genotype *hs-FLP; AP-1-GFP; act>>GAL4, UAS-RFP/ T(2:3)SM6A, TM6B, Tb* were crossed to flies bearing *UAS-GluR1; tubGAL80$^{ts}$*. Larvae were cultured at 18 °C and heat shocked in a 37 °C water bath for 10 min at 82 hr AEL, returned to 18 °C, and upshifted to 30 °C for 20 hr at 168 h AEL, followed by dissection and immunostaining. *tubGAL80$^{ts}$* flies were used as a non-ablating control.

### Regeneration scoring and wing measurements

Wings of adult flies from heat shocked larvae were scored and measured after genotype blinding by another researcher. Scoring was performed on anesthetized adults by binning into a regeneration scoring category (*Harris et al., 2020*; *Klemm et al., 2021*). Wing measurements were performed by removing wings, mounting in Permount solution (Thermo Fisher Scientific) and imaged using a Zeiss Discovery.V8 microscope. Wing area was measured using the Fiji software. Male and female adults

were measured separately to account for sex differences in wing size using a reproducible measuring protocol that excludes the variable hinge region of the wing (details of measuring protocol available on request). Statistics were performed using GraphPad Prism 10.0.

## Immunohistochemistry

Larvae were dissected in 1 x PBS followed by a 20 min fix in 4% paraformaldehyde in PBS (PFA). After 3 washes in 0.1% PBST (1 x PBS +0.1% Triton-X), larvae were washed in 0.3% PBST and then blocked in 0.1% PBST with 5% normal goat serum (NGS) for 30 min. Primary staining was done overnight at 4 °C, and secondary staining was done for 4 hr at room temperature. The following primary antibodies were obtained from the Developmental Studies Hybridoma Bank: mouse anti-Nubbin (1:25), mouse anti-Wg (1:100), mouse anti-Mmp1 C-terminus (1:100), mouse anti-Mmp1 catalytic domain (1:100), mouse anti-LacZ (1:100), mouse anti-discs large (1:50), mouse anti-yH2Av (1:100), and rat anti-DE-cadherin (1:100). Rabbit anti-cDcp-1 (1:1000), mouse anti-PH3 (1:500), and rabbit anti-HA (1:1000) were obtained from Cell Signaling Technologies. Rat anti-Zfh-2 was generously gifted by Chris Doe. Anti-rabbit 647, anti-rat 647, anti-mouse 555, and anti-mouse 488 secondary antibodies were obtained from Invitrogen and used at a 1:500 dilution. DAPI (1:1000) was used as a counterstain. Images were obtained on a Zeiss AxioImager.M2 with ApoTome and a Leica TCS SP8 LCSM (NIH SIG award 1 S10 OD023691-01) housed in the Regenerative Medicine Imaging Facility at Arizona State University. For each experiment at least 15 discs were analyzed prior to choosing a representative image, and each experiment was replicated as a biological repeat (arising from an independent genetic cross). Sample sizes were determined based on phenotype consistency and standards in the field. Images were processed using Affinity Photo.

## EdU staining, TUNEL assay, and DHE staining

### EdU

The Click-It EdU Alexa Fluor 555 Imaging Kit (Invitrogen C10338) was used to assay cell proliferation. Briefly, imaginal discs were dissected and labeled with 1 µl EdU in 1 ml PBS for 20 min, fixed in 4% PFA for 20 min and immunolabeled (as necessary), followed by a 30 min Click-It reaction that was performed as directed in the EdU manual.

### TUNEL

The TUNEL assay was performed with the ApopTag Red In Situ Apoptosis Detection Kit (Millipore S7165). Dissected larvae were fixed in 4% paraformaldehyde, followed by a 10 min wash in 75 ml equilibration buffer. Discs were then submerged in 55 µl working strength TdT enzyme for 3 hr at 37 °C. The reaction was stopped by adding 1 ml stop/wash buffer and incubating for 10 min at room temperature, followed by three washes in PBS. Immunolabeling was performed by incubating the tissue preps with 65 µl of anti-digoxigenin rhodamine overnight at room temperature.

### Dihydroethidium (DHE)

DHE labeling was performed by incubating freshly dissected wing imaginal discs in Schneider's Media with 1 µl of 10 mM DHE reconstituted in 1 ml DMSO (for a working concentration of 10 µm DHE) for 10 min, followed by three 5 min washes in PBS and immediately mounting and imaging.

## Quantification and statistical analysis

Adult wings, mean fluorescence intensity, and cell counts were measured using Fiji. GraphPad Prism 10.0 was used for statistical analysis and graphical representation. Graphs depict the mean of each treatment, while error bars represent the standard deviation. The mean fluorescence intensity of EdU labeling was quantified in Fiji. In each experiment, the mean fluorescence intensity (MFI) of the wing pouch was normalized to the MFI of the entire disc. The mean and standard deviation for each normalized treatment was calculated and used for statistical analysis. The sample size and P values for all statistical analyses are indicated in the figure legends. Statistical significance was evaluated in Prism 10.0 using a Student's T-test or a one-way ANOVA with a multiple comparisons test.

## Acknowledgements

The authors would like to thank Dr. Tin Tin Su of UC Boulder, Dr. Chris Doe of the University of Oregon, and Dr. Tian Xie of the Stowers Institute for their generous gift of stocks and reagents. We thank the current members of the Harris lab for useful input and feedback. We thank the Bloomington *Drosophila* Stock Center and Developmental Studies Hybridoma Bank for stocks and reagents. This work was supported by a grant from the Eunice Kennedy Shriver National Institute of Child Health and Human Development (NICHD) R21HD102765 and the National Institute of General Medical Sciences (NIGMS) R01GM147615 to Robin Harris.

## Additional information

### Funding

| Funder | Grant reference number | Author |
|---|---|---|
| Eunice Kennedy Shriver National Institute of Child Health and Human Development | R21HD102765 | Robin E Harris |
| National Institute of General Medical Sciences | R01GM147615 | Robin E Harris |

The funders had no role in study design, data collection and interpretation, or the decision to submit the work for publication.

### Author contributions

Jacob W Klemm, Conceptualization, Data curation, Formal analysis, Validation, Investigation, Visualization, Methodology, Writing – original draft, Writing – review and editing; Chloe Van Hazel, Data curation, Investigation, Methodology; Robin E Harris, Conceptualization, Resources, Data curation, Formal analysis, Supervision, Funding acquisition, Writing – original draft, Project administration, Writing – review and editing

### Author ORCIDs

Jacob W Klemm ⓘ http://orcid.org/0000-0002-1414-5089
Chloe Van Hazel ⓘ http://orcid.org/0009-0007-6175-2678
Robin E Harris ⓘ https://orcid.org/0000-0001-6945-6741

Reviewer #2 (Public review): https://doi.org/10.7554/eLife.101114.3.sa1
Reviewer #3 (Public review): https://doi.org/10.7554/eLife.101114.3.sa2
Author response https://doi.org/10.7554/eLife.101114.3.sa3

## Additional files

### Supplementary files

Supplementary file 1. Detailed genotypes of fly stocks used in experiments of the indicated figure and panel.

MDAR checklist

### Data availability

All data generated or analysed during this study are included in the manuscript and supporting files; complete genotypes have been provided for each figure in the Supplementary Genotypes file.

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
