## [Editor Report · eLife Assessment]

This manuscript reports **fundamental** discoveries on how necrotic cells contribute to organ regeneration through apoptotic signalling to produce cells with non-lethal apoptotic caspase activity that contribute to the regenerated tissue. These findings will be of broad interest to those who study wound repair and tissue regeneration. The strength of the evidence is **solid** and has been improved in the revised version.

---

## [Referee Report · Reviewer #2 (Public review)]

In this revised manuscript, Klemm et al., build on top of past published findings (Klemm et al., 2021) to characterize caspase activation in distal cells following necrotic tissue damage within the *Drosophila* wing imaginal disc. Previously in Klemm et al., 2021, the authors describe necrosis-induced-apoptosis (NiA) following the development of a genetic system to study necrosis that is caused by the expression of a constitutive active GluR1 (Glutamate/Ca2+ channel), and they discovered that the appearance of NiA cells were important for promoting regeneration.

In this manuscript, the authors investigate how tissues regenerate following necrotic cell death. They find that:

(1) the cells of the wing pouch are more likely to have non-autonomous caspase activation than other regions within the wing imaginal disc (hinge and notum),

(2) two signaling pathways that are known to be upregulated during regeneration, Wnt (wingless) and JAK/Stat signaling, act to prevent additional NiA in pouch cells, and may partially explain the region specificity,

(3) the presence of NiA (and/or NiCP) cells promotes regenerative proliferation in the late stages of regeneration,

(4) not all caspase-positive cells are cleared from the epithelium (these cells are then referred to as Necrosis-induced Caspase Positive (NiCP) cells), these NiCP cells continue to live and promote proliferation in adjacent cells,

(5) the initiator caspase Dronc is important for creating NiA/NiCP cells and for these cells to promote proliferation. Animals heterozygous for a Dronc null allele show a decrease in regeneration following necrotic tissue damage. In the revised manuscript, the authors provide improvements through additional data quantifications and text changes to better explain NiA/NiCP lineage tracing methods.

The study has the potential to be broadly interesting due to the insights into how tissues differentially respond to necrosis as compared to apoptosis to promote regeneration. The paper raises many interesting questions for future investigation, including what is the nature of the signaling between the damaged tissue and the NiA/NiCP responsive areas (such as the identity of the DAMPs)? What determines if these cells at a distance undergo apoptosis or remain viable in the tissue as caspase-positive cells? And since the authors have data that indicates that the phenomenon is distinct from 'undead cells', what are the mechanisms by which these cells promote local proliferation?

---

## [Referee Report · Reviewer #3 (Public review)]

The manuscript "Regeneration following tissue necrosis is mediated by non-apoptotic caspase activity" by Klemm et al. is an exploration of what happens to a group of cells that experience caspase activation after necrosis occurs some distance away from the cells of interest. These experiments have been conducted in the *Drosophila* wing imaginal disc, which has been used extensively to study the response of a developing epithelium to damage and stress. The authors revise and refine their earlier discovery of apoptosis initiated by necrosis, here showing that many of those presumed apoptotic cells do not complete apoptosis. Thus, the most interesting aspect of the paper is the characterization of a group of cells that experience mild caspase activation in response to an unknown signal, followed by some effector caspase activation and DNA damage, but that then recover from the DNA damage, avoid apoptosis, and proliferate instead.

The authors have addressed the concerns raised, including those about drawing conclusions from RNAi knockdown without evaluating the efficacy of the knockdown, and in doing so they revised their conclusions after ascertaining that the Zfh2 RNAi was not effective.

The authors have added quantification of the imaging data throughout, which strengthens their conclusions.

In addition, the authors have revised some of the text describing the changes in EdU signal and added explanations of reagents such as the caspase sensors to clarify the experimental approaches, results, and interpretation of those results.

The authors have also addressed the minor concerns and questions about the figures and text.

A few questions remain, which the authors may choose to address.

(1) The hh>Stat92ERNAi was assessed by the 10xSTAT-GFP reporter, as shown in Fig 2 Supp1 F. The authors point out the marked reduction in GFP in the ventral part of the hinge but do not comment on the lack of change in GFP in the dorsal part of the hinge. However, the open arrowhead in Figure 2H indicating the lack of cDcp-1 signal in the hinge in the same experiment points to the dorsal hinge, where the reporter suggests no difference in JAK-STAT signaling.

(2) The data used to conclude that DRONC-DN and UAS-DIAP1 do not affect regenerative proliferation were normalized EdU intensities. As discussed in the prior review round, normalized EdU may not be a good comparison across experimental conditions given that the remainder of the disc may also have altered EdU incorporation, so this measurement may not be enough by itself to draw conclusions about regenerative proliferation. To strengthen the conclusion that regenerative proliferation is unaffected under these conditions, the authors may want to consider using a second measure such as adult wing size, PCNA, or quantitate mitoses via anti-phospho histone H3 staining.

---

## [Author Response]

The following is the authors’ response to the original reviews.

**Reviewer #1 (Public Review):**

**Summary:**
In previous work, the authors described necrosis-induced apoptosis (NiA) as a consequence of induced necrosis. Specifically, experimentally induced necrosis in the distal pouch of larval wing imaginal discs triggers NiA in the lateral pouch. In this manuscript, the authors confirmed this observation and found that while necrosis can kill all areas of the disc, NiA is limited to the pouch and to some extent to the notum, but is excluded from the hinge region. Interestingly and unexpectedly, signaling by the Jak/Stat and Wg pathways inhibits NiA. Further characterization of NiA by the authors reveals that NiA also triggers regenerative proliferation which can last up to 64 hours following necrosis induction. This regenerative response to necrosis is significantly stronger compared to discs ablated by apoptosis. Furthermore, the regenerative proliferation induced by necrosis is dependent on the apoptotic pathway because RNAi targeting the RHG genes is sufficient to block proliferation. However, NiA does not promote proliferation through the previously described apoptosis-induced proliferation (AiP) pathway, although cells at the wound edge undergo AiP. Further examination of the caspase levels in NiA cells allowed the authors to group these cells into two clusters: some cells (NiA) undergo apoptosis and are removed, while others referred to as Necrosis-induced Caspase Positive (NiCP) cells survive despite caspase activity. It is the NiCP cells that repair cellular damage including DNA damage and that promote regenerative proliferation. Caspase sensors demonstrate that both groups of cells have initiator caspase activity, while only the NiA cells contain effector caspase activity. Under certain conditions, the authors were also able to visualize effector caspase activity in NiCP cells, but the level was low, likely below the threshold for apoptosis. Finally, the authors found that loss of the initiator caspase Dronc blocks regenerative proliferation, while inhibiting effector caspases by expression of p35 does not, suggesting that Dronc can induce regenerative proliferation following necrosis in a non- apoptotic manner. This last finding is very interesting as it implies that Dronc can induce proliferation in at least two ways in addition to its requirement in AiP.Strengths:This is a very interesting manuscript. The authors demonstrate that epithelial tissue that contains a significant number of necrotic cells is able to regenerate. This regenerative response is dependent on the apoptotic pathway which is induced at a distance from the necrotic cells. Although regenerative proliferation following necrosis requires the initiator caspase Dronc, Dronc does not induce a classical AiP response for this type of regenerative response. In future work, it will be very interesting to dissect this regenerative response pathway genetically.Weaknesses:No weaknesses were identified.

We thank the reviewer for their positive evaluation and kind words.

**Reviewer #2 (Public Review):**
Summary / Strengths:In this manuscript, Klemm et al., build on past published findings (Klemm et al., 2021) to characterize caspase activation in distal cells following necrotic tissue damage within the *Drosophila* wing imaginal disc. Previously in Klemm et al., 2021, the authors describe necrosis-induced-apoptosis (NiA) following the development of a genetic system to study necrosis that is caused by the expression of a constitutive active GluR1 (Glutamate/Ca2+ channel), and they discovered that the appearance of NiA cells were important for promoting regeneration.In this manuscript, the authors aim to investigate how tissues regenerate following necrotic cell death. They find that the cells of the wing pouch are more likely to have non-autonomous caspase activation than other regions within the wing imaginal disc (hinge and notum),two signaling pathways that are known to be upregulated during regeneration, Wnt (wingless) and JAK/Stat signaling, act to prevent additional NiA in pouch cells, and may explain the region specificity, the presence of NiA cells promotes regenerative proliferation in late stages of regeneration, not all caspase-positive cells are cleared from the epithelium (these cells are then referred to as Necrosis-induced Caspase Positive (NiCP) cells), these NiCP cells continue to live and promote proliferation in adjacent cells, the caspase Dronc is important for creating NiA/NiCP cells and for these cells to promote proliferation. Animals heterozygous for a Dronc null allele show a decrease in regeneration following necrotic tissue damage.The study has the potential to be broadly interesting due to the insights into how tissues differentially respond to necrosis as compared to apoptosis to promote regeneration.Weaknesses:However, here are some of my current concerns for the manuscript in its current version:The presence of cells with activated caspase that don't die (NiCP cells) is an interesting biological phenomenon but is not described until Figure 5. How does the existence of NiCP cells impact the earlier findings presented? Is late proliferation due to NiA, NiCP, or both? Does Wg and JAK/STAT signaling act to prevent the formation of both NiA and NiCP cells or only NiA cells? Moreover, the authors are able to specifically manipulate the wound edge (WE) and lateral pouch cells (LP), but don't show how these manipulations within these distinct populations impact regeneration. The authors provide evidence that driving UAS-mir(RHG) throughout the pouch, in the LP or the WE all decrease the amount of NiA/NiCP in Figure 3G-O, but no data on final regenerative outcomes for these manipulations is presented (such as those presented for Dronc-/+ in Fig 7M). The manuscript would be greatly enhanced by quantification of more of the findings, especially in describing if the specific manipulations that impacted NiA /NiCP cells disrupt end-point regeneration phenotypes.

We have added a line to the results to clarify that we believe the finding that some NiA likely persist as NiCP does not affect our conclusions up to this point.

We have added a statement emphasizing the results from our first paper, which demonstrate that LP>miRHG expression reduces the overall capacity to regenerate.

Quantification of the change in posterior NiA number have been added to Figure 2L to strengthen the evidence. Likewise, we have included quantification of the E2F time course presented in Figure 3A (Figure 3 – Figure supplement 1C), and quantification of the change in GC3Ai signal over time has been added to Figure 5 - Figure supplement 1D to emphasize the perdurance of GC3Ai-positive NiA/NiCP.

How fast does apoptosis take within the wing disc epithelium? How many of the caspase(+) cells are present for the whole 48 hours of regeneration? Are new cells also induced to activate caspase during this time window? The author presented a number of interesting experiments characterizing the NiCP cells. For the caspase sensor GC3Ai experiments in Figure 5, is there a way to differentiate between cells that have maintained fluorescent CG3Ai from cells that have newly activated caspase? What is the timeline for when NiA and NiCP are specified? In addition, what fraction of NiCP cells contribute to the regenerated epithelium? Additional information about the temporal dynamics of NiA and NiCP specification/commitment would be greatly appreciated.

We have included more information concerning the kinetics of apoptotic cell removal, and how this compares to the observations we have made with NiA/NiCP in our GC3Ai experiments. Additionally, we have included a quantification of the percent of the whole wing pouch with GC3Ai signal over time (Figure 5F) as well as the distal wing pouch with GC3Ai signal over time (Figure 5 – Figure supplement 1D) to further support the idea that NiCP persist over time.

We acknowledge that our GC3Ai time course unfortunately cannot confirm whether the increase in GC3Ai signal over time is due to cells with new caspase activity or proliferating NiCP and have included this point in the discussion.

We attempted to track the lineage of NiA/NiCP into the pupal and adult wings with CasExpress and DBS, however the results of these experiments were inconsistent, and therefore we did not feel confident to include these data or draw conclusions in either direction. We are currently designing variations of these lineage trace tools in order to better track the lineage of these cells that we hope to include in a future paper.

The notum also does not express developmental JAK/STAT, yet little NiA was observed within the notum. Do the authors have any additional insights into the differential response between the pouch and notum? What makes the pouch unique? Are NiA/NiCP cells created within other imaginal discs and other tissues? Are they similarly important for regenerative responses in other contexts?

We have added a brief mention of these points to the appropriate results section to avoid further increasing the length of the discussion.

Data on the necrosis of other imaginal discs through FLP/FRT clone formation in haltere and leg discs has been added to Figure 1 Figure supplement 1J, and described in the text.

**Reviewer #3 (Public Review):**
The manuscript "Regeneration following tissue necrosis is mediated by non- apoptotic caspase activity" by Klemm et al. is an exploration of what happens to a group of cells that experience caspase activation after necrosis occurs some distance away from the cells of interest. These experiments have been conducted in the *Drosophila* wing imaginal disc, which has been used extensively to study the response of a developing epithelium to damage and stress. The authors revise and refine their earlier discovery of apoptosis initiated by necrosis, here showing that many of those presumed apoptotic cells do not complete apoptosis. Thus, the most interesting aspect of the paper is the characterization of a group of cells that experience mild caspase activation in response to an unknown signal, followed by some effector caspase activation and DNA damage, but that then recover from the DNA damage, avoid apoptosis, and proliferate instead. Many questions remain unanswered, including the signal that stimulates the mild caspase activation, and the mechanism through which this activation stimulates enhanced proliferation.The authors should consider answering additional questions, clarifying some points, and making some minor corrections:Major concerns affecting the interpretation of experimental results:Expression of STAT92E RNAi had no apparent effect on the ability of hinge cells to undergo NiA, leading the authors to conclude that other protective signals must exist. However, the authors have not shown that this STAT92E RNAi is capable of eliminating JAK/STAT signaling in the hinge under these experimental conditions. Using a reporter for JAK/STAT signaling, such as the STAT-GFP, as a readout would confirm the reduction or elimination of signaling. This confirmation would be necessary to support the negative result as presented.

We have included data demonstrating our ability to knock down JAK/STAT activity in the hinge with UAS-Stat92E^RNAi^ (Figure 2 – Figure supplement 1E and F). Additionally, we have included a quantification of posterior NiA/NiCP with the Stat92E^RNAi^ (as well as wg^RNAi^ and Zfh-2^RNAi^, Figure 2L) to strengthen our conclusion that JAK/STAT and WNT signaling acts to regulate NiA formation within the pouch.

Similarly, the authors should confirm that the Zfh2 RNAi is reducing or eliminating Zfh2 levels in the hinge under these experimental conditions, before concluding that Zfh2 does not play a role in stopping hinge cells from undergoing NiA.

We have repeated this experiment with a longer knockdown using a GAL4 driver that expresses from early larval stages until our evaluation at L3, but were unable to demonstrate a loss of Zfh-2 with IF labeling. Additionally, we have quantified posterior NiA/NiCP with a Zfh-2RNAi (Figure 2L) and do find a slight increase in NiA/NiCP number, however this change is not significant. We have altered our conclusions to reflect these new data.

EdU incorporation was quantified by measuring the fluorescence intensity of the pouch and normalizing it to the fluorescence intensity of the whole disc. However, the images show that EdU fluorescence intensity of other regions of the disc, especially the notum, varied substantially when comparing the different genetic backgrounds (for example, note the substantially reduced EdU in the notum of Figure 3 B' and B'). Indeed, it has been shown that tissue damage can lead to suppression of proliferation in the notum and elsewhere in the disc, unless the signaling that induces the suppression is altered. Therefore, the normalization may be skewing the results because the notum EdU is not consistent across samples, possibly because the damage-induced suppression of proliferation in the notum is different across the different genetic backgrounds.

To more accurately reflect the observations that we have made with the EdU assay, we have changed our terminology to indicate that the EdU signal is more localized to the damaged tissue in ablated discs, thus taking into account the relative changes across the disc, rather than referring to it as an increase in the pouch. To further strengthen our observation that damage results in a localized proliferation, we have included a quantification of the E2F time course presented in Figure 3A (Figure 3 – Figure supplement 1C), which underscores the trend observed in our EdU experiments.

The authors expressed p35 to attempt to generate "undead cells". They take an absence of mitogen secretion or increased proliferation as evidence that undead cells were not generated. However, there could be undead cells that do not stimulate proliferation non-autonomously, which could be detected by the persistence of caspase activity in cells that do not complete apoptosis. Indeed, expressing p35 and observing sustained effector caspase activation could help answer the later question of what percentage of this cell population would otherwise complete apoptosis (NiA, rescued by p35) vs reverse course and proliferate (NiCP, unaffected by p35).

In our previous work, we showed that P35 expression impairs our ability to detect effector caspases with IF-based tools. This can also be seen in Figure 4 of this work (Figure 4C and F). Given that P35 expression precludes our ability to label and assay effector caspase activity visually, and thus address the concerns outlined above, we relied on other tools such as reporters of AiP mitogens (*wg-lacZ & dpp-lacZ*) to assay whether NiA participate in AiP. As a functional readout, we also paired P35 expression with the EdU assay to test whether proliferation was altered by the presence of undead cells. The results discussed in Figure 4 lead us to conclude that NiA likely do not participate in the canonical AiP feedforward loop, although it is possible that these experiments generate another type of undead cell – one that utilizes a different mechanism to promote proliferation.

It is unclear if the authors' model is that the NiCP cells lead to autonomous or non-autonomous cell proliferation, or both. Could the lineage-tracing experiments and/or the experiments marking mitosis relative to caspase activity answer this question?

We have added further details to the discussion on the potential for NiA/NiCP to induce cell autonomous/non-autonomous proliferation.

Many of the conclusions rely on single images. Quantification of many samples should be included wherever possible.

We have added quantification to strengthen the results of Figures 2, 3 and 5.

Why does the reduction of Dronc appear to affect regenerative growth in females but not males?

We have repeated this regeneration scoring experiments and have increased the N for control versus *droncI29* mutant males, however the results of the analysis for male wing size remain not significant, although the general trend that *droncI29* wings are slightly smaller. While there could be sex-specific differences in the capacity to regenerate that contribute to this observation, it is unclear what the underlying mechanism could be.

**Reviewer #1 (Recommendations for the authors):**
The work in this paper is already very complete and very well worked out. The conclusions are well supported by the data in this manuscript. I do not have any experimental requests, only a few minor and formal requests/questions.(1) Why does Diap1 overexpression not affect regenerative proliferation, whereas mir(RHG) and dronc[I29] do, given that Diap1 acts between RHG and Dronc?

We speculate on this point in the discussion section but have adjusted some of the phrasing for clarity.

(2) I assume that the authors used the cleaved Dcp-1 antibody from Cell Signaling Technologies. I recommend that the authors refer to this antibody as cDcp-1 in text and figures as this antibody specifically detects the cleaved, and thus activated form of Dcp-1, and not the uncleaved, inactive form of Dcp-1 which has a uniform expression in the discs.

Changed to cDcp-1.

(3) Line 299: Hay et al. 1994 did not show that p35 inhibits Drice and Dcp-1 (in fact, both genes were not even cloned yet). This was shown by Meier et al. 2000 and Hawkins et al. 2000. Please correct references.

Corrected.

(4) Line 574/575. Meier et al. 2000 did not show that Dronc is mono-ubiquitylated. This was shown by Kamber-Kaya et al., 2017. Please correct.

Corrected.

**Reviewer #2 (Recommendations for the authors):**
(1) Does domeless knockdown cause apoptosis without tissue ablation (Figures 2C-E)? Currently, the non-ablation control is not shown.

Domeless knockdown does not cause apoptosis in the absence of ablation (Added Figure 2 – Figure supplement 1A).

(2) The supplemental experiment with zfh2-RNAi is hard to interpret because there is no evidence of RNAi knockdown based on the staining with the anti-Zfh2 antibody.

As noted above, a longer *zfh-2* knockdown does not appear to alter Zfh-2 protein levels. A quantification of posterior NiA/NiCP following knockdown shows a slight (non-significant) increase in posterior NiA/NiCP. Considering these new results, we have altered our interpretation within the appropriate results and discussion sections.

(3) The authors should consider adding a diagram showing where mir(RHG) and DIAP1 are in the apoptotic/caspase activation pathway (Figure 7N).

Completed, Figure 7N and 7O.

**Reviewer #3 (Recommendations for the authors):**
(1) Figure 2 I -The purported increase in NiA should be quantitated relative to the NiA in G across many discs.

Completed (Figure 2L)

(2) Figure 2 M - contrary to the conclusion drawn, the posterior Dcp1 does not appear different from that in the control (K). This conclusion that the NiA does not occur in the margin could be better supported with more images/quantification.

We have exchanged the image for a representative one that more clearly shows the lack of margin NiA and highlighted with an arrowhead (Figure 2K)

(3) Figure 2 supp 1 E - the "slight increase" in NiA in the pouch is relative to which control? Can this conclusion be supported by quantification?

Figure 2L now quantifies this change.

(4) Figure 2 Supp 1 D, E - these discs supposedly have Zfh2 RNAi expressed, but there appears to be no reduction in Zfh2.

We were unable to demonstrate a reduction of Zfh2, even with a longer knockdown. Considering these new data, we have altered our conclusions from the Zfh2 experiments.

(5) Figure 2 Supp 1 I - please quantitate the Dcp-1 across many discs to support the conclusion.

This is the UAS-wg experiment, which we decided to remove from the quantification given the non-specific increase in cDcp-1 throughout the disc (likely as a result from ectopic Wg expression).

(6) Figure 4 legend M - The authors conclude that the experiment indicates that "NiA promote proliferation independent of AiP". It would be more precise to say that NiA cells do not secrete AiP mitogens and do not increase the proliferation of surrounding cells when prevented from completing apoptosis. To say that the NiA-induced proliferation does not require AiP would require eliminating AiP, perhaps through reaper hid grim knockdown or mitogen knockdown.

Corrected.

Minor concerns and clarification needed:(7) Line 61 - consider the distinction between a feed-forward loop and a positive feedback loop.

Corrected.

(8) Line 338 - it would be helpful to have a brief explanation of what the GC3Ai consists of and how it reports caspase activity.

Corrected.

(9) Line 343 - the authors should clarify by what they mean when they state GC3Ai-positive cells are "associated with" mitotic cells. Are the GC3Ai cells undergoing mitosis? Or is the increase in mitosis non-autonomous?

Adjusted. “associated with adjacent proliferative cells”.

(10) Lines 392-394 - the authors should add brief descriptions of how the Drice-Based sensor and the CasExpress function, so the readers can better understand the distinctions between these sensors and the previously mentioned sensors (anti-Dcp1 and GC3Ai). In addition, please clarify how the Gal80ts modulates the sensitivity of the CasExpress.

Descriptions of DBS and CasExpress and additional clarification provided.

(11) Line 413: How does Gal80ts suppress the background developmental caspase signal, and how does this suppression lead to NiCP cells expressing GFP?

This section has been reworded to clarify.

(12) Line 417 - which GFP label is referred to here?

This section has been reworded to clarify.

(13) Line 445 is the first mention of the CARD domain - it could be introduced more fully and explained why the DroncDN's lack of effect on proliferation excludes the CARD domain as being important.

Clarified. See also the discussion for the significance of the CARD domain as dispensable for regenerative proliferation following necrosis.

(14) Line 452 - "As mentioned" - the manuscript has not previously mentioned DIAP1 modification of the CARD domain and what that modification does. Perhaps the previous explanatory text was inadvertently removed?

Corrected.

(15) The Discussion is a lengthy list of experiments that the authors did not do or observations they were unable to make. This section could benefit from a more in-depth discussion of necrosis and the possibility that NiCP cells contribute to repair after injury across contexts and species.

We have made several changes to the discussion that elaborate on some of the points listed in the public reviews.

(16) All figures: Consider making single-channel panels grayscale to aid visualization. Also consider using color combinations that can be distinguished by color-blind readers.

We appreciate these suggestions and will consider them for future manuscripts.

(17) All figure legends - are error bars SD or SEM?

Standard deviation. Added to appropriate legends.

(18) Figure 1A,C - it would be helpful in the diagrams to note when the necrosis occurs/completes.

The endpoint of necrosis is not well defined, given the simultaneous changes that occur with regeneration. Thus, we opted to not include an indicator of when necrotic ablation ends.

(19) Figure 1B - it would be helpful to name the GAL4 drivers whose expression domain is depicted to correlate with the terms used in the text.

Completed.

(20) Figure 1 legend- what do the different colors of the arrowheads denote? The dotted lines are in R' and S', not N' and O'.

Completed.

(21) Figure 2G - the yellow dashed line is not in the same place in the two images.

Corrected.

(22) Figure 2I - what is the open arrowhead?

Completed (Figure 2I legend).

(23) Figure 3 legend - please describe what the time course is observing (EdU).

Completed.

(24) Figure 4 - please include the yellow boxes in the Dcp-1 channels.

Completed.

(25) Figure 5 F' - add the arrowheads to all the panels. The yellow arrowhead appears to be pointing to nothing.

Completed.

(27) Figure 5 legend - what is a "cytoplasmic undisturbed cell"? What is the arrowhead in G? J and J' should show the same view at different time points or different views at the same time point.

Figure legend has been corrected.

(28) Figure 5 Supp 1 would be especially helped by having more single-channel panels in grayscale.

For clarity and consistency, we chose to maintain the different color channels.

(29) Figure 5 Supp 1 D and E - It would be helpful to have higher magnification and arrows pointing to the cells of interest. Why are there TUNEL+ cells that do not have caspase activation (green)?

We have added arrowheads as suggested. We believe the disparity in TUNEL and GC3Ai signals are a result of the different sensitivities of the IF staining and the TUNEL assay.

(30) Figure 5 Supp 1 F - perhaps the arrowheads should be in all panels - they point to empty spaces with no H2Av staining in the final panel. Perhaps a higher magnification image would make the "strong overlap" of the two signals more apparent?

We have added arrowheads where appropriate.

(31) Figure 6 D-E - does the widespread GFP lineage tracing signal suggest that most cells in the repaired tissue originated from cells that once had caspases activity?

Possibly, however given that CasExpress leads to significant developmental labeling, we were unable to determine to what extent the signal in this experiment comes from NiA/NiCP activity versus developmental labeling. Note that tubGAL80ts is not present in this experiment.

(32) Writing corrections:Line 343 "positive" is misspelled.

Completed

Line 429 - a word may be missing.

Completed

Line 639 - the word "day" may be missing.

Completed

Line 658 - what temperature was the recovery?

Completed

Lines 706-708 - were the discs incubated in 55 mL and 65 mL of liquid, or a smaller volume?

Completed